# Stochastic bond dynamics facilitates alignment of malaria parasite at erythrocyte membrane upon invasion

**Sebastian Hillringhaus†, Anil K Dasanna†, Gerhard Gompper\*, Dmitry A Fedosov\***

Theoretical Physics of Living Matter, Institute of Biological Information Processing and Institute for Advanced Simulation, Forschungszentrum Jülich, Jülich, Germany

**Abstract** Malaria parasites invade healthy red blood cells (RBCs) during the blood stage of the disease. Even though parasites initially adhere to RBCs with a random orientation, they need to align their apex toward the membrane in order to start the invasion process. Using hydrodynamic simulations of a RBC and parasite, where both interact through discrete stochastic bonds, we show that parasite alignment is governed by the combination of RBC membrane deformability and dynamics of adhesion bonds. The stochastic nature of bond-based interactions facilitates a diffusive-like re-orientation of the parasite at the RBC membrane, while RBC deformation aids in the establishment of apex-membrane contact through partial parasite wrapping by the membrane. This bond-based model for parasite adhesion quantitatively captures alignment times measured experimentally and demonstrates that alignment times increase drastically with increasing rigidity of the RBC membrane. Our results suggest that the alignment process is mediated simply by passive parasite adhesion.

**\*For correspondence:**
g.gompper@fz-juelich.de (GG);
d.fedosov@fz-juelich.de (DAF)

†These authors contributed equally to this work

**Competing interests:** The authors declare that no competing interests exist.

## Introduction

Malaria is a dangerous mosquito-borne disease which kills nearly 0.5 million of people every year (*World Health Organisation, 2018*). It is caused by a protozoan parasite of the genus Plasmodium and proceeds in several stages (*Miller et al., 2002*; *Cowman et al., 2012*; *White et al., 2014*). After about 10 days from the initial infection through a mosquito bite, an infected liver releases a large number of merozoites, egg-shaped parasites with a typical size of $1 - 2\mu m$ (*Bannister et al., 1986*; *Dasgupta et al., 2014*), into the blood stream. The blood stage of malaria infection is a clinically relevant stage, where merozoites invade healthy red blood cells (RBCs) and multiply inside by utilizing the RBC internal resources. This intra-erythrocytic development is essential for merozoites to be hidden from the immune system and avoid clearance. After about 48 hours post RBC invasion, infected RBCs are ruptured and new merozoites are released into the blood stream to repeat this reproduction cycle. Thus, RBC invasion by merozoites is crucial not only for parasite survival, but also for further multiplication.

RBC invasion by merozoites is preceded by three key events: (i) initial attachment, (ii) re-orientation or alignment of the parasite such that its apex is facing the RBC membrane, and (iii) formation of a tight junction (*Koch and Baum, 2016*). The apex contains all required machinery to invade RBCs after the tight junction is formed (*Cowman and Crabb, 2006*). At physiological hematocrit levels with a volume fraction of RBCs close to 40%, initial attachment of merozoites can be considered almost immediate after their egress from infected RBCs. However, the initial attachment has a random parasite orientation, which rarely provides direct alignment of the apex toward the membrane required to start the invasion. This implies that the parasite alignment is an extremely crucial step for successful invasion, which needs to be completed within a couple of minutes, as after this time period merozoites generally lose their ability to invade RBCs (*Crick et al., 2014*). To facilitate

parasite alignment, merozoites contain a surface coat of proteins, mainly GPI-anchored, which can bind to the RBC membrane (*Bannister et al., 1986*; *Gilson et al., 2006*; *Beeson et al., 2016*). However, one of the main difficulties in the investigation of RBC-parasite interactions is that exact receptor-ligand bindings remain largely unknown. Electron microscopy images (*Bannister et al., 1986*) of merozoites adhered to a RBC suggest that along with short bonds of length $\simeq 20\,nm$, connecting the two cells, there exist much longer bonds of lengths up to $150\,nm$, which may play an important role in early stages of merozoite adhesion to the RBC membrane. Furthermore, these long bonds have a much lower density than short bonds. Even though adhesion kinetics of such bonds remain unknown, recent optical tweezers experiments (*Crick et al., 2014*) indicate the adhesion force of spent merozoites to the RBC membrane to be within the range of 10 to 40pN.

Another important aspect during merozoite alignment is the deformation of the RBC membrane. Dynamic membrane deformations of various magnitudes are often observed (*Dvorak et al., 1975*; *Gilson and Crabb, 2009*; *Glushakova et al., 2005*; *Crick et al., 2013*) and are thought to aid in the alignment process (*Weiss et al., 2015*; *Hillringhaus et al., 2019*). Recent live-cell imaging experiments show a positive correlation between RBC deformations and eventual merozoite alignment (*Weiss et al., 2015*). Most merozoites that successfully invade RBCs induce considerable membrane deformations, while the invasion success is much less frequent without preceding RBC deformations. Furthermore, these experiments lead to an estimate of an average alignment time of about $16\,s$ (*Weiss et al., 2015*). A recent simulation study by *Hillringhaus et al., 2019*, with RBC-parasite adhesion modeled by a homogeneous interaction potential, has confirmed the importance of membrane deformations, which facilitate parasite alignment through its partial wrapping by the membrane. However, this model shows static (not dynamic) membrane deformations and leads to average alignment times of less than $1\,s$, indicating that an essential aspect of the alignment process has not been captured. Another speculation is that dynamic membrane deformations are induced actively by merozoites through changing locally the concentration of Ca+ ions (*Lew and Tiffert, 2007*; *McCallum-Deighton and Holder, 1992*). This proposition has been confronted by recent experiments (*Introini et al., 2018*), which show that calcium release by parasite starts only at the invasion stage. Therefore, RBC membrane deformations are potentially induced by a passive mechanism, such as parasite adhesion.

In this paper, we focus on the passive compliance hypothesis (*Introini et al., 2018*) which assumes that RBC deformations and parasite alignment result from parasite adhesion interactions rather than from some active mechanism. Thus, our central question is whether parasite alignment can be explained purely by the passive compliance hypothesis. In contrast to the recent simulation study by *Hillringhaus et al., 2019*, where RBC-parasite interactions are represented by a laterally smooth potential, the adhesion model presented here is based on discrete stochastic bonds between parasite and RBC membrane. This is a key step toward a realistic description of RBC-merozoite adhesion, since it eliminates the major shortcomings of the previous potential-based model such as unrealistically fast alignment times and the absence of dynamic membrane deformations. Even though receptor-ligand interactions which determine parasite alignment are largely not known, our bond-based interaction model still incorporates a few experimental details such as the range of adhesion interactions and density of different agonists (*Bannister et al., 1986*). In particular, bonds of different lengths, that is long and short two-state bond interactions, are employed in the model. The bond-based parasite adhesion model generates an erratic motion of the parasite at the RBC membrane, visually similar to that observed experimentally (*Weiss et al., 2015*). Furthermore, it results in alignment times which agree quantitatively with those measured in experiments (*Weiss et al., 2015*; *Yahata et al., 2012*) and confirms the importance of membrane deformations for successful parasite alignment. The model is also used to investigate the effect of various adhesion parameters, such as bond extensional rigidities and kinetic rates, and ligand densities, on the parasite alignment process. Future investigations with this model can consider more realistic scenarios such as parasite adhesion and alignment under blood flow conditions.

The article is organized as follows. First, we introduce and calibrate our hydrodynamic model, where simulation parameters are tuned to quantitatively match several characteristics of the parasite motion at the RBC membrane from available experimental data by *Weiss et al., 2015*. Then, RBC membrane deformations and alignment times are investigated for this reference parameter set and several cases of altered bond kinetics and rigidities, and ligand densities. Finally, the effect of membrane stiffness on alignment times is studied.

## Results

The RBC membrane is modeled as a network of $N_{\mathrm{rbc}} = 3000$ vertices that are distributed uniformly on the membrane surface and connected by $N_s$ springs (*Gompper and Kroll, 2004*; *Fedosov et al., 2010a*; *Fedosov et al., 2010b*; *Fedosov et al., 2014*). Our RBC membrane model incorporates elastic and bending resistance, and its biconcave shape is obtained by constraining the total surface area and enclosed volume of the membrane. Similar to the RBC, a parasite is modeled by $N_{\mathrm{para}} = 1230$ vertices distributed homogeneously on its surface. The egg-like shape of a merozoite (see *Figure 1a*) is approximated as (*Dasgupta et al., 2014*; *Hillringhaus et al., 2019*)

$$\left(r_x^2 + r_y^2 + r_z^2\right)^2 = (R_a - R_b)r_x\left(r_y^2 + r_z^2\right) + R_a r_x^3, \tag{1}$$

where $R_a = 1.5\,\mu m$ and $R_b = 1.05\,\mu m$ are diameters along the major and minor axes of the parasite, respectively. The parasite is much less deformable than the RBC, as no deformations of parasite body are visible in experiments (*Weiss et al., 2015*; *Crick et al., 2014*). Therefore, the merozoite is considered to be a rigid body, whose dynamics can be described by equations involving force and torque on the parasite's center of mass and directional vector (*Heard, 2006*).

Both RBC and parasite are immersed in a fluid and the hydrodynamic interactions are modeled by the dissipative particle dynamics (DPD) method (*Hoogerbrugge and Koelman, 1992*; *Español and Warren, 1995*). The interaction of parasite and RBC membrane has two components. The first component corresponds to an excluded-volume repulsion to prevent an overlap between the two cells, which is modeled by the repulsive part of the Lennard-Jones (LJ) potential with a minimum possible distance $\sigma = 0.2\,\mu m$. The distance $\sigma$ can be considered as an effective membrane thickness of a surface constructed from overlapping spheres with a diameter $\sigma$. Generally, $\sigma$ depends on the resolution length of both the RBC membrane and parasite (about $0.2\,\mu m$ in our models) and is chosen large enough to guarantee no artificial membrane intersection or overlap between the cells. The effect of the precise value of $\sigma$ on simulation results is expected to be small and will be discussed later. The second interaction component represents adhesion which is modeled by discrete dynamic bonds between RBC and parasite vertices. Each parasite vertex represents one of the two different types of ligands: (i) long ligands with an effective binding range $\ell_{\mathrm{eff}}^{\mathrm{long}} = 100\,\mathrm{nm}$ and (ii) short ligands with an effective binding range $\ell_{\mathrm{eff}}^{\mathrm{short}} = 20\,\mathrm{nm}$. Both ligand types are distributed randomly at the parasite surface with fixed ligand densities $\rho_{\mathrm{long}}$ and $\rho_{\mathrm{short}}$, such that their sum $\rho_{\mathrm{long}} + \rho_{\mathrm{short}}$ is equal to the parasite vertex density $\rho_{\mathrm{para}}$. Receptors for ligand binding are modeled by RBC vertices, each of which can bind only a single ligand, irrespective of its type. Due to the effective membrane thickness characterized by $\sigma$, long and short bonds can be formed by

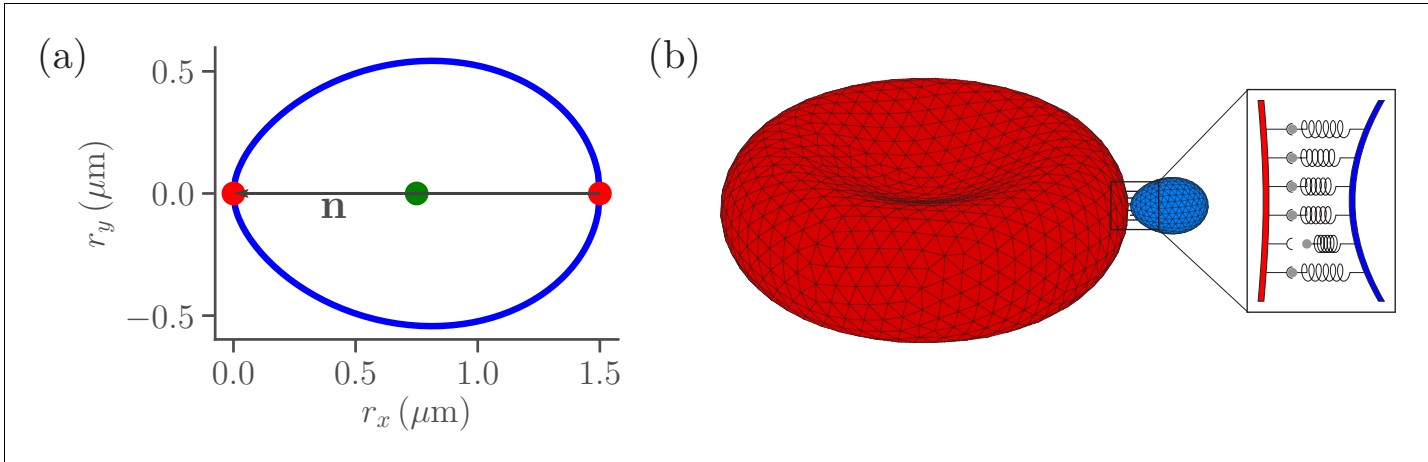

**Figure 1.** Sketch of parasite and RBC models. (**a**) Two-dimensional sketch of a parasite with a directional vector **n** from the parasite's back at $r_x = 1.5\,\mu m$ to its apex at $r_x = 0$. (**b**) Three-dimensional triangulated surfaces of a RBC (red) and a parasite (blue). Bonds between the parasite and RBC can form within the contact zone which is illustrated by a magnified view, where discrete receptor-ligand interactions (or bonds) are sketched. A bond can form with a constant on-rate $k_{\mathrm{on}}$ and break with a constant off-rate $k_{\mathrm{off}}$.

**Table 1.** Simulation parameters given in both model and physical units.

The effective RBC diameter $D_0 = \sqrt{A_0/\pi}$ sets a basic length, the thermal energy $k_{\mathrm{B}}T$ defines an energy scale, and RBC relaxation time $\tau = \eta D_0^3/\kappa$ sets a time scale in the simulated system, where $A_0$ is the RBC surface area, $\kappa$ is the bending rigidity, and $\eta$ is the fluid dynamic viscosity. The values of bending rigidity $\kappa$, 2D shear $\mu$ and Young's $Y$ moduli are chosen such that they correspond to average properties of a healthy RBC. Parameters $\sigma$ and $\epsilon$ correspond to RBC-parasite excluded-volume interactions represented by the purely repulsive LJ potential in **Equation 11**.

| Parameter | Simulation value | Physical value |
|---|---|---|
| $A_0$ | 133.5 | $133.5\,\mu\mathrm{m}^2$ |
| $D_0$ | $\sqrt{A_0/\pi} = 6.5$ | $6.5\,\mu\mathrm{m}$ |
| $k_{\mathrm{B}}T$ | 0.01 | $4.282 \times 10^{-21}\,\mathrm{J}$ |
| $\tau$ | $\eta D_0^3/\kappa = 725.8$ | $0.92\,\mathrm{s}$ |
| $\eta$ | 1.85 | $1 \times 10^{-3}\,\mathrm{Pa\,s}$ |
| $\kappa$ | $70\,k_{\mathrm{B}}T$ | $3.0 \times 10^{-19}\,\mathrm{J}$ |
| $\mu$ | $4.6 \times 10^4\,k_{\mathrm{B}}T/D_0^2$ | $4.8\,\mu\mathrm{N/m}$ |
| $Y$ | $1.82 \times 10^5\,k_{\mathrm{B}}T/D_0^2$ | $18.9\,\mu\mathrm{N/m}$ |
| $N_{\mathrm{para}}$ | 1230 | |
| $N_{\mathrm{rbc}}$ | 3000 | |
| $\sigma$ | $0.031\,D_0$ | $0.2\,\mu\mathrm{m}$ |
| $\epsilon$ | $1000\,k_{\mathrm{B}}T$ | $4.282 \times 10^{-18}\,\mathrm{J}$ |

bound long and short ligands if the distance between RBC and parasite vertices is smaller than $\ell_0 + \ell_{\mathrm{eff}}^{\mathrm{long}}$ and $\ell_0 + \ell_{\mathrm{eff}}^{\mathrm{short}}$, respectively, where $\ell_0 = 2^{1/6}\sigma$ is the equilibrium spring length that corresponds to the cutoff of repulsive interactions. Note that existing bonds are allowed to stretch beyond their effective binding ranges, see section 'Methods and models' for more details.

To relate simulation units to physical units, a basic length scale is defined as the effective RBC diameter $D_0 = \sqrt{A_0/\pi}$ ($A_0$ is the membrane area), an energy scale as $k_{\mathrm{B}}T$, and a time scale as RBC membrane relaxation time $\tau = \eta D_0^3/\kappa$, where $\eta$ is the fluid viscosity and $\kappa$ is the bending rigidity of

**Table 2.** List of bond parameters that are used to calibrate displacement of the parasite at the RBC membrane in simulations (see **Video 1**) against available experimental data (**Weiss et al., 2015**), as shown in **Figure 2b**.

The parameter values in simulations are given in terms of the length scale $D_0$, energy scale $k_{\mathrm{B}}T$, and timescale $\tau = \eta D_0^3/\kappa$. The densities of long and short ligands are given in terms of parasite vertex density $\rho_{\mathrm{para}} \simeq 270\,\mu m^{-2}$. Note that $\rho_{\mathrm{long}} + \rho_{\mathrm{short}} = \rho_{\mathrm{para}}$ in all simulations.

| Parameter | Simulation value | Physical value |
|---|---|---|
| $\ell_{\mathrm{eff}}^{\mathrm{long}}$ | $0.0154\,D_0$ | $100\,\mathrm{nm}$ |
| $\ell_{\mathrm{eff}}^{\mathrm{short}}$ | $0.0031\,D_0$ | $20\,\mathrm{nm}$ |
| $\rho_{\mathrm{long}}$ | $0.4\,\rho_{\mathrm{para}}$ | $107\,\mu m^{-2}$ |
| $\rho_{\mathrm{short}}$ | $0.6\,\rho_{\mathrm{para}}$ | $161\,\mu m^{-2}$ |
| $k_{\mathrm{on}}^{\mathrm{long}}$ | $36.3\,\tau^{-1}$ | $39.6\,\mathrm{s}^{-1}$ |
| $k_{\mathrm{on}}^{\mathrm{short}}$ | $290.3\,\tau^{-1}$ | $317.0\,\mathrm{s}^{-1}$ |
| $k_{\mathrm{off}}$ | $72.58\,\tau^{-1}$ | $79.2\,\mathrm{s}^{-1}$ |
| $\lambda_{\mathrm{long}}$ | $25.3 \times 10^5\,k_{\mathrm{B}}T/D_0^2$ | $0.264\,\mathrm{pN/nm}$ |
| $\lambda_{\mathrm{short}}$ | $8.45 \times 10^5\,k_{\mathrm{B}}T/D_0^2$ | $0.0882\,\mathrm{pN/nm}$ |

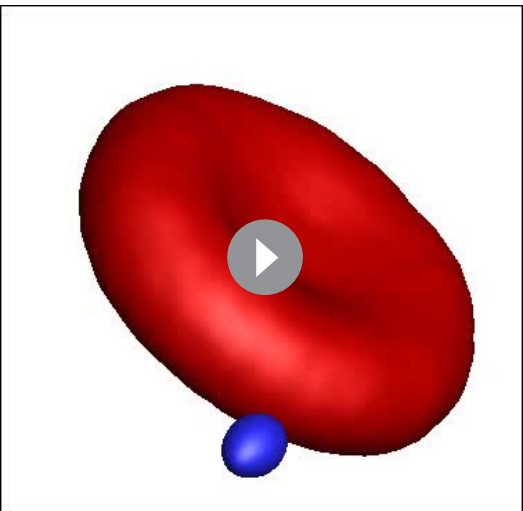

**Video 1.** Parasite motion at the membrane of a deformable RBC for the reference RBC-parasite interactions from **Table 2**. $k_{\text{off}}/k_{\text{on}}^{\text{long}} = 2$. See **Figure 2a**.
https://elifesciences.org/articles/56500#video1

the membrane. All simulation parameters in model and physical units are given in **Tables 1** and **2**. Average properties of a healthy RBC correspond to $D_0 \simeq 6.5\,\mu\text{m}$ with $A_0 = 133.5\,\mu\text{m}^2$ (**Evans and Skalak, 1980**) and $\tau \approx 0.92\,\text{s}$ for $\kappa = 3 \times 10^{-19}\,\text{J}$ (**Evans, 1983**; **Fedosov et al., 2010a**) and $\eta = 1\,\text{mPa}\,\text{s}$.

To better understand the effect of various adhesion properties on parasite alignment, several parameters such as bond formation and rupture rates, bond rigidity, and ligand densities are varied. For each fixed parameter set, a number of simulations are performed and the results are combined and/or averaged, which is necessary due to the stochastic nature of bond-based interaction as well as thermal fluctuation effects within the fluid. Note that each simulation is performed for a different random choice of parasite vertices which represent long and short ligands, while their densities remain fixed, see section 'Methods and models'.

## Calibration of RBC-parasite interactions

A parasite adhered to the RBC membrane exhibits visually an irregular diffusive-like motion observed experimentally (**Weiss et al., 2015**), which is controlled by the ligand densities $\rho_{\text{long}}$ and $\rho_{\text{short}}$, bond rigidities $\lambda_{\text{long}}$ and $\lambda_{\text{short}}$, and the bond formation ($k_{\text{on}}^{\text{long}}$, $k_{\text{on}}^{\text{short}}$) and rupture ($k_{\text{off}}$) rates that are currently not known. Nevertheless, available experiments (**Bannister et al., 1986**) suggest that the number of short bonds in RBC-merozoite interaction is lager than the number of long bonds, which is reflected in the ligand densities $\rho_{\text{long}}$ and $\rho_{\text{short}}$ assumed for our parasite model (see **Table 2**). To calibrate RBC-parasite interactions, parasite dynamics at the RBC membrane (see **Video 1**) is quantified by its fixed-time displacement, which is measured by tracking the distance $\Delta d$ traveled by the parasite at fixed intervals of time $\Delta t$, see **Figure 2a**. Particle tracking is employed to measure $\Delta d$ from available experiments (**Weiss et al., 2015**), where $\Delta t$ is selected to be $1\,\text{s}$, which is the time resolution of the experimental videos. Only time ranges, within which parasites remain visible and the RBC is not moving much, are included in the analysis.

**Figure 2b** compares experimental and simulated characteristics of fixed-time displacements for the interaction parameters given in **Table 2**. This set of parameters (further referred to as reference case) is obtained by varying $\rho_{\text{long}}$, $\rho_{\text{short}}$, $\lambda_{\text{long}}$, $\lambda_{\text{short}}$, $k_{\text{on}}^{\text{short}}$, $k_{\text{on}}^{\text{long}}$, and $k_{\text{off}}$ until a good agreement between experimental and simulated parasite displacements is reached. However, the effective binding ranges of long and short ligands remain fixed at $\ell_{\text{eff}}^{\text{long}} = 100\,nm$ and $\ell_{\text{eff}}^{\text{short}} = 20\,nm$ in this calibration procedure. The variance of experimental displacements in **Figure 2b** is larger than that in simulations due to a limited sample size of experimental data (20 samples). Note that this set of parameters is likely not unique, and other combinations of the parameters, which result in statistically similar parasite-displacement characteristics, can probably be found.

To further characterize the parasite motion on the RBC membrane, the mean-squared displacement (MSD) of the parasite's center of mass is computed in simulations and shown in **Figure 2c**. At long enough times $t \gtrsim 3\,\text{s}$, the parasite exhibits diffusive-like motion, indicated by a linear increase of the MSD curve with time. For shorter timescales, the MSD of parasite motion shows a transient anomalous subdiffusion, which may occur, for instance, in the case of sticky particle dynamics with alterations between sticking (i.e., stopping its motion for some time) and diffusing states (**Saxton, 2007**; **Höfling and Franosch, 2013**). The transient sticky dynamics is an appropriate description for an adhered parasite, where sticking periods correspond to time intervals within which no bonds are formed or ruptured. The diffusive-like dynamics is governed by the number of bonds $n_b$ and their

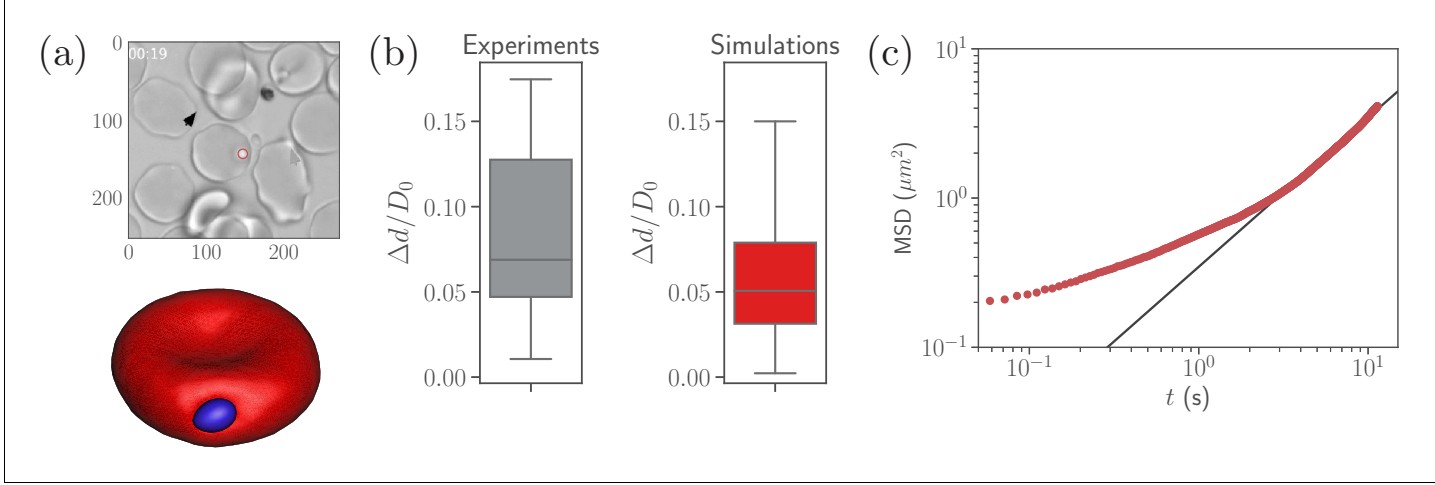

**Figure 2.** Calibration of parasite adhesion parameters. (a) A time instance of parasite motion at RBC membrane from an experimental video (*Weiss et al., 2015*) (top) and simulation (bottom), see also *Video 1*. To obtain the distribution of merozoite fixed-time displacements, the marked parasite (red circle) is tracked over the course of its interaction with the RBC membrane. (b) Comparison between experimental (20 samples) and simulated (100 samples) fixed-time displacements ($\Delta d$) of the parasite at RBC membrane, which is normalized by the effective RBC diameter $D_0 = \sqrt{A_0/\pi}$ calculated from the membrane area $A_0$. By adapting the interaction parameters, the displacement distribution in simulations is calibrated against the experimental distribution. The resulting reference parameters for our model can be found in *Table 2*. (c) Mean squared displacement (MSD) of a parasite from simulations as a function of time. The black solid line marks a diffusive regime with $\mathrm{MSD} \sim t$. Note the subdiffusive dynamics for short times, less than about $1\,\mathrm{s}$.

The online version of this article includes the following source data for figure 2:

**Source data 1.** Source data for graphs shown in *Figure 2(b,c)*.

on- and off-rates, as an adhered particle becomes slower and eventually gets arrested when $n_b$ is increased and the rates are decreased (*Jana and Mognetti, 2019*).

## Parasite alignment

Recent experiments suggest that a successful RBC invasion strongly correlates not only with the distance between parasite apex and RBC membrane, but also with a perpendicular alignment of the merozoite toward the cell membrane (*Koch and Baum, 2016*). Furthermore, the junctional (invasion initiating) interaction range $r_{\mathrm{junc}}$ of the parasite's apex is known to be around $10\,\mathrm{nm}$ (*Bannister et al., 1986*). Based on these observations, we define two quantities, (i) the apex distance $d_{\mathrm{apex}}$ from the RBC membrane, and (ii) the alignment angle $\theta$ that characterizes parasite orientation, both sketched in *Figure 3a*. Here, $d_{\mathrm{apex}}$ is defined as the distance between the parasite apex and the nearest membrane vertex,

$$d_{\mathrm{apex}} = \min_i \left( \left| \mathbf{r}_{\mathrm{apex}} - \mathbf{r}_i \right| \right), \tag{2}$$

the alignment angle $\theta$ as the angle between the parasite's directional vector $\mathbf{n}$ and the normal $\mathbf{n}^{\mathrm{face}}$ of a triangular face whose center is closest to the apex,

$$\theta = \arccos\left( \mathbf{n} \cdot \mathbf{n}^{\mathrm{face}} \right). \tag{3}$$

*Figure 3b,c* shows distributions of apex distance $d_{\mathrm{apex}}$ and alignment angle $\theta$ for the calibrated RBC-parasite interactions. Both characteristics are represented by distributions as the merozoite is very dynamic at the membrane surface. Minimum values of $d_{\mathrm{apex}}$ in *Figure 3b* correspond to the parasite's apex being very close to the membrane (i.e., $d_{\mathrm{apex}} \approx \sigma$), whereas maximum values generally represent a configuration where the parasite is adhered sideways to the RBC. Furthermore, low values of $\theta$ in *Figure 3c* characterize the sideways adhesion orientation, while large values of $\theta$ represent a good alignment configuration. Note that an ideal merozoite alignment would be achieved if $d_{\mathrm{apex}}$ is less than $\sigma + r_{\mathrm{junc}}$ ($r_{\mathrm{junc}} = 10\,\mathrm{nm}$) and the alignment angle is $\theta \approx \pi$. Due to a discrete

representation of the membrane, perfect alignment is unlikely, which requires to slightly relax these conditions. Therefore, we define a successful parasite alignment by the criteria

$$d_{\text{apex}} \leq 2^{1/6}\sigma + r_{\text{junc}} \;\&\; \theta \geq 0.8\pi. \tag{4}$$

The choice of $0.8\pi$ in *Equation 4* is also partially driven by the RBC discretization length of about $0.2\,\mu\text{m}$. Half circumference of the parasite corresponds to $\pi R_a/2 = 2.36\,\mu m$, which is about twelve RBC discretization lengths. This means that our resolution in determining angle $\theta$ is close to $0.1\pi$, so that the window of $0.2\pi$ in the alignment criteria is large enough to avoid strong discretization effects.

In experiments, merozoite alignment times are measured as time intervals between initial parasite adhesion and the beginning of invasion (*Weiss et al., 2015*). Similarly, alignment time in simulations is calculated as the time required for the parasite to meet the alignment criteria in *Equation 4* starting from an initial adhesion contact (i.e., formation of a few bonds). *Figure 4b* presents a distribution of alignment times from 86 statistically independent DPD simulations for the reference RBC-parasite interactions in *Table 2*. The alignment times range between $1\,\text{s}$ and $26\,\text{s}$ with an average value of $9.53\,\text{s}$. For comparison, the average alignment time was reported to be $16\,\text{s}$ by *Weiss et al., 2015*, and the range of alignment times between $7\,\text{s}$ and $44\,\text{s}$ was found by *Yahata et al., 2012*, which agree reasonably well with our model predictions. Differences in alignment times between simulations and experiments are possibly due to a limited experimental statistics (e.g. only 10 samples in the study by *Yahata et al., 2012*) and/or selected model parameters, as the distribution of alignment times in our model can be altered by changing RBC-parasite interactions. Therefore, further experiments and possible model improvements are needed to clarify the source of existing differences.

Note that the sample size (about 100) in simulations is limited by the computational cost. A single simulation, corresponding to a total physical time of about $26\,\text{s}$, requires approximately 168 core hours on the supercomputer JURECA (*Jülich Supercomputing Centre, 2018*) at Forschungszentrum Jülich. Therefore, a direct brute-force approach for the investigation of the effect of various parameters on the parasite alignment time is not feasible. To overcome this problem, Monte-Carlo (MC) sampling (see section 'Methods and models' for details), which is based on a two-dimensional probability map of parasite alignment characteristics ($d_{\text{apex}}$, $\theta$) illustrated in *Figure 4a*, is employed to determine the differences in alignment times for various parameter sets. Such a probability map is computed from several direct DPD simulations of RBC-parasite adhesive interactions. Then, the MC procedure is used to model stochastic jumps between neighboring alignment states ($d_{\text{apex}}^i$, $\theta^j$) within the probability map, starting from a randomly selected initial state and continuing until the alignment criteria in *Equation 4* are met, and the number of MC steps represents the alignment time. Distribution of alignment times $t_n$ from the MC sampling is shown in *Figure 4c* for the reference parameter set. Clearly, the distributions obtained by direct (*Figure 4b*) and MC (*Figure 4c*) simulations are very similar, verifying the reliability of the MC approach. Note that alignment times $t_n$ from MC sampling are measured in terms of MC steps, since MC simulations do not have an intrinsic time-scale. The average alignment time for the reference parameter set is denoted as $\langle t_{n,\text{ref}} \rangle$ and assumed to be equivalent to $9.53\,\text{s}$, the average alignment time from direct DPD simulations of RBC-parasite adhesion. This implies that $10^4$ MC steps correspond to about $15\,\text{s}$.

## Membrane deformation and parasite dynamics

A recent simulation study by *Hillringhaus et al., 2019* with a laterally homogeneous adhesion potential has demonstrated that the deformation of RBC membrane is crucial for a successful parasite alignment. Further, we show that ligand density, bond rigidity and kinetics not only control the parasite motion at the membrane surface, but also directly affect membrane deformation. To quantify the strength of membrane deformations, a change in total energy between the deformed state and the equilibrium state of the RBC membrane is computed as (*Hillringhaus et al., 2019*)

$$\Delta E_{\text{rbc}} = E_{\text{rbc}}^{\text{deform}} - E_{\text{rbc}}^{\text{equil}}. \tag{5}$$

*Figure 5* shows temporal changes in deformation energy, number of bonds, head distance, and alignment angle for the reference case. Two major contributions to the deformation energy (i.e.

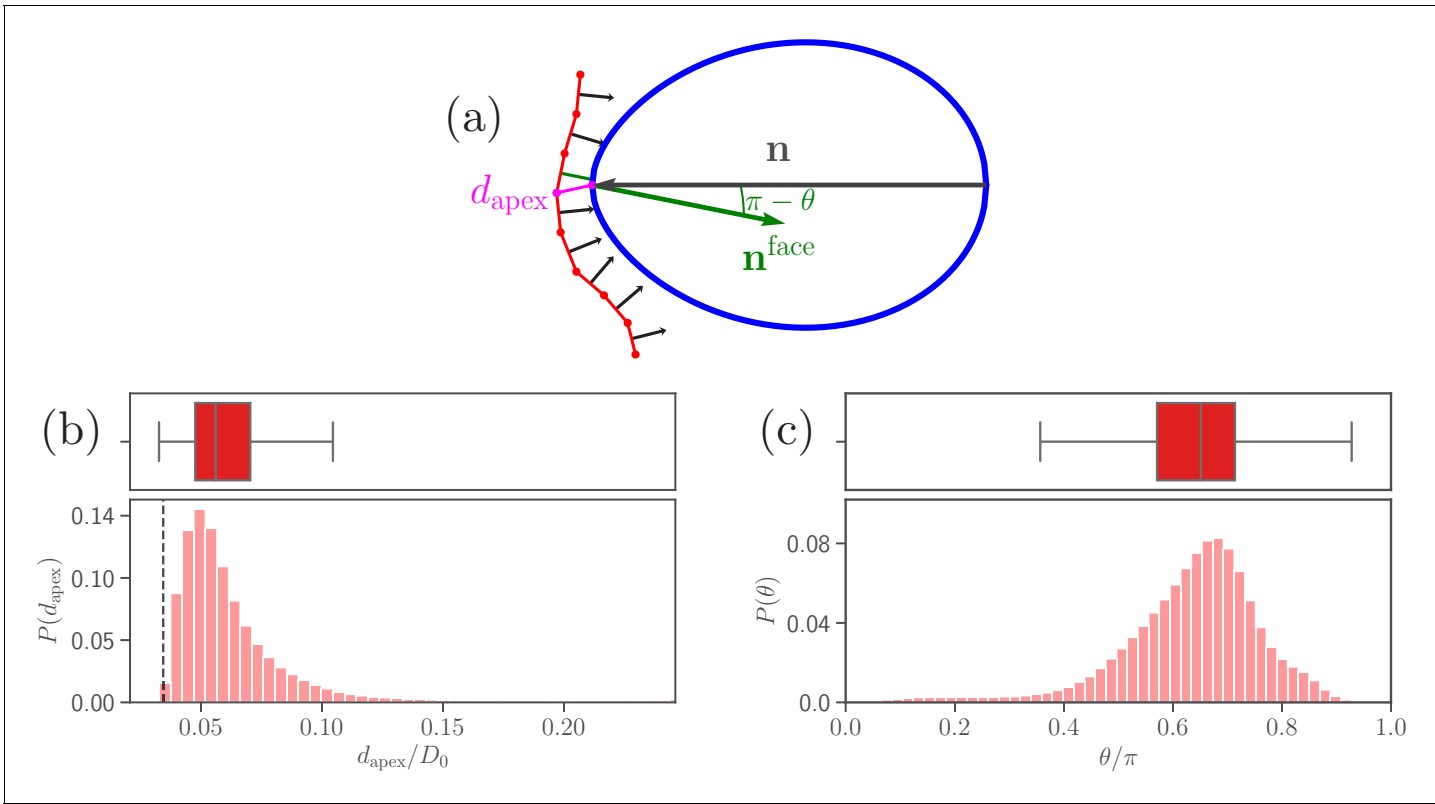

**Figure 3.** Parasite adhesion to a deformable RBC. (**a**) Sketch of apex distance $d_{\text{apex}}$ and alignment angle $\theta$. The apex distance $d_{\text{apex}}$ is defined as a distance (magenta line) between the parasite's apex and the closest vertex of RBC membrane. The alignment angle $\theta$ corresponds to the angle between the parasite's directional vector (black arrow) and the normal vector $\mathbf{n}^{\text{face}}$ (green arrow) of a triangular face whose center is closest to the apex. Note that the angle $\pi - \theta$ is drawn in the plot. (**b** and **c**) Probability distributions of the apex distance $d_{\text{apex}}/D_0$ and the alignment angle $\theta/\pi$. Data are obtained for parameters shown in *Table 2*, and accumulated starting from an initial adhesion contact (i.e., formation of a few bonds). The dashed line in the apex distance distribution indicates the cutoff $2^{1/6}\sigma$ of repulsive LJ interactions. Note that a good parasite alignment requires small values of $d_{\text{apex}}/D_0$ and values of $\theta/\pi$ close to unity.

The online version of this article includes the following source data and figure supplement(s) for figure 3:

**Source data 1.** Source data for graphs shown in *Figure 3b,c* and *Figure 3—figure supplement 1*.
**Figure supplement 1.** Parasite adhesion to a rigid RBC (see section 'Effect of RBC rigidity').

elastic stretching $\Delta E_{\text{sp}}$ and bending $\Delta E_{\text{bend}}$ energies) indicate that membrane deformation is very dynamic and has a strong variability in its intensity. This is due to the dynamic formation and dissociation of long and short bonds between the merozoite and RBC membrane.

An interesting observation is that the head distance and alignment angle in *Figure 5* fluctuate around some average values, indicating that the parasite has a preferred orientation, which is consistent with a peak in the probability map in *Figure 4a*. To assess whether the most likely values of $d_{\text{apex}}$ and $\theta$ are mainly determined by the egg-like parasite shape, or also depend on the mechanical properties of the membrane, $d_{\text{apex}}$ and $\theta$ distributions in *Figure 3* for a deformable RBC are compared with those for the parasite adhered to a rigidified membrane (see section 'Effect of RBC rigidity') in *Figure 3—figure supplement 1*. Clearly, in the case of a rigid membrane, the preferred $d_{\text{apex}}$ and $\theta$ values are determined by the egg-like parasite shape, corresponding to a configuration with maximum adhesion area. In comparison to the deformable membrane (*Figure 3*), the peak in $d_{\text{apex}}$ for the rigid RBC (see *Figure 3—figure supplement 1*) is shifted further away from zero. This indicates that the degree of wrapping has a significant effect on the preferred values of $d_{\text{apex}}$ and $\theta$. Therefore, in addition to the egg-like parasite shape, RBC membrane properties, such as bending rigidity, shear elasticity, and local curvature, affect the most probable values of $d_{\text{apex}}$ and $\theta$. Furthermore, the fluctuations of $d_{\text{apex}}$ and $\theta$ from their average values in *Figure 5* represent parasite motion toward its apex or bottom due to stochastic bond dynamics. Thus, the parasite dynamics at the

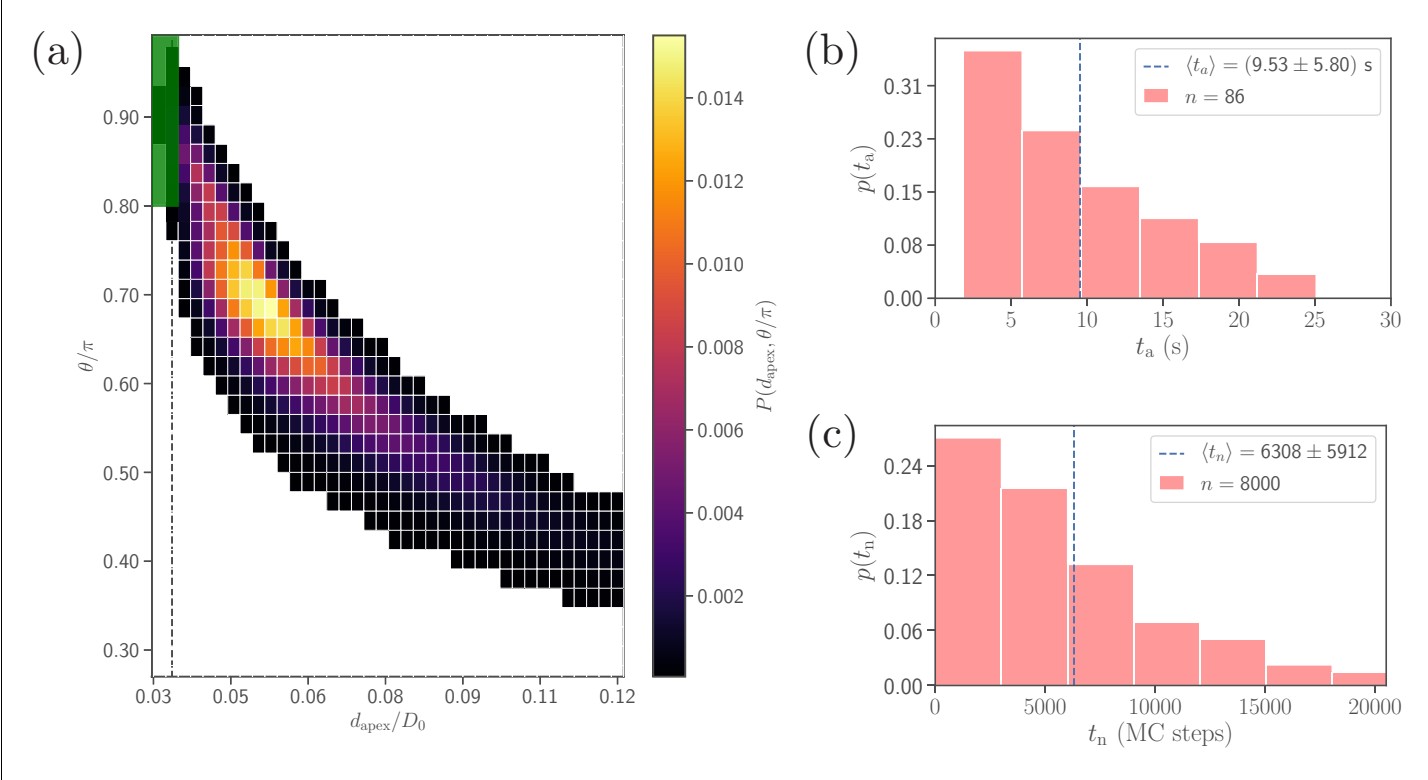

**Figure 4.** Comparison of alignment times obtained from direct DPD simulations and MC sampling. (a) Two-dimensional probability map as a function of $d_{apex}$ and $\theta$. Each bin represents a single alignment state and the color corresponds to probability of that state. The dark green area ($d_{apex}/D_0 \leq 0.036$ and $\theta/\pi \geq 0.8$, compare with *Equation 4*) represents the criteria for a successful alignment. The black dashed line corresponds to the cutoff $2^{1/6}\sigma$ of repulsive LJ interactions. (b) Distribution of alignment times $t_a$ obtained from 86 statistically independent DPD simulations. $t_a$ is defined as a time interval starting from an initial adhesive contact (i.e., formation of a few bonds) to the instance when the alignment criteria for $d_{apex}$ and $\theta$ in *Equation 4* are met. The average alignment time is equal to $\langle t_a \rangle \simeq 9.53 \sim$ s. (c) Alignment time distribution from MC sampling using the probability map in (a). The alignment time is defined as a number of MC steps needed to satisfy the alignment criteria, as the MC procedure does not have an inherit timescale. Note that the sample size in MC modeling (8000 trajectories) is much larger than that in (b).

The online version of this article includes the following source data for figure 4:

**Source data 1.** Source data for graphs shown in *Figure 4a–c*.

membrane can be described as a superposition of the rolling motion around its directional vector with a preferred orientation and intermediate fluctuations of parasite orientation toward its apex or the bottom. The rotational motion around the directional vector is preferred because it is not associated with a significant energy cost, while fluctuations in the orientation toward the merozoite's apex or bottom have an energy penalty.

A further noteworthy result from simulations is that a successful alignment occurs more frequently in the concave areas of RBC dimples than at the convex rim of the membrane. This is due to the fact that the cell dimples have a favorable local curvature or a lower energy penalty for membrane wrapping (*Agudo-Canalejo and Lipowsky, 2015*; *Yu et al., 2018*), which leads to a stronger parasite wrapping by the membrane, and thus a larger probability for successful alignment. *Figure 5—figure supplement 1* shows that the merozoite forms more bonds in the dimples than at the RBC rim, confirming the position-dependent differences in membrane wrapping. Furthermore, our simulations show that merozoites move frequently into the dimple areas, starting from the initial rim contact, and remain there for the majority of simulation time. This behavior is again due to a more energetically favorable adhesion position within RBC dimples in comparison to the RBC rim. Energetically favorable parasite wrapping within the RBC dimples might be also advantageous for the subsequent entry into the cell.

The dynamic adhesive behavior of the parasite in the current stochastic bond-based model is in striking contrast to the previous adhesion model (*Hillringhaus et al., 2019*) based on a

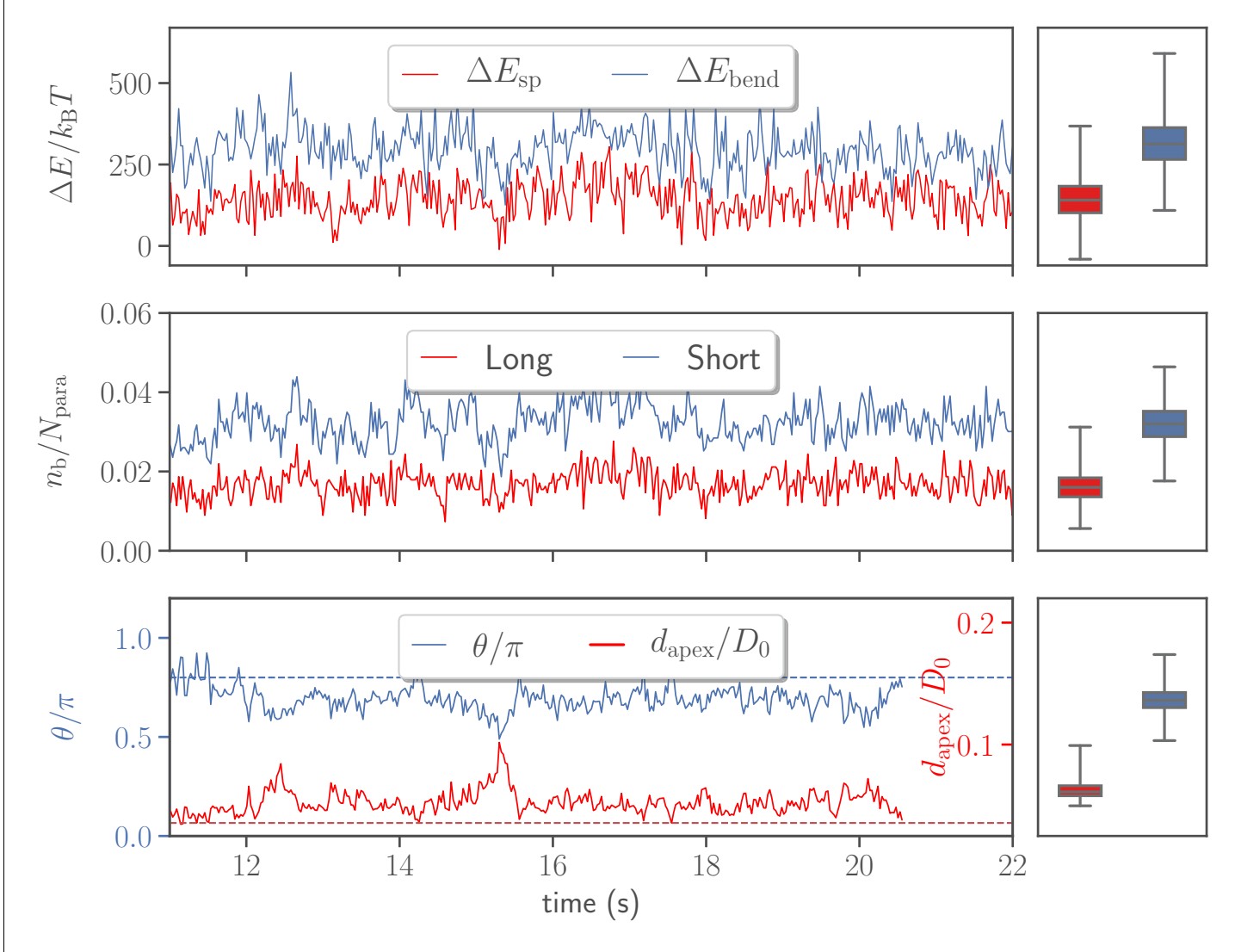

**Figure 5.** Variations in stretching $\Delta E_{sp}$ and bending $\Delta E_{bend}$ energies, the number of bonds $n_b$, the head distance $d_{apex}$, and the alignment angle $\theta$ as a function of time for the default parameter set given in *Table 2*. Temporal changes in the number of bonds are shown for both long and short bond types. The dashed lines in the bottom plot correspond to the alignment criteria in *Equation 4*. For all quantities, the corresponding averages and variances represented by box plots are depicted on the right.

The online version of this article includes the following source data and figure supplement(s) for figure 5:

**Source data 1.** Source data for graphs shown in *Figure 5* and *Figure 5—figure supplements 1* and *2*.

**Figure supplement 1.** Dependence of parasite wrapping on the position at RBC membrane.

**Figure supplement 2.** Different alignment characteristics, including (**a**) deformation energy, (**b**) number of bonds, (**c**) apex distance, (**d**) alignment angle, and (**e**) fixed-time displacement, for several values of parameter $\sigma$ which determines the effective membrane thickness.

homogeneous interaction potential between the two cells, where no dynamic deformations were observed. A qualitative correspondence between these two models can be understood by considering a ratio $k_{on}/k_{off} = \exp(\Delta U_b/k_B T)$, where $\Delta U_b$ is the binding energy of a single bond (*Bell, 1978*; *Schwarz and Safran, 2013*). Thus, the ratio $k_{on}/k_{off}$ directly controls the average number of bonds $<n_b>$ and the strength of adhesion (see section 'Effect of bond properties on parasite alignment'), which are correlated with RBC deformation energy $\Delta E_{rbc}$. Similarly, in the parasite adhesion model with a homogeneous interaction potential (*Hillringhaus et al., 2019*), the strength of adhesion potential controls membrane deformations. Even though average membrane deformations can be compared for these two models, the stochastic bond-based adhesion model results in a very

different diffusive-like dynamics of the parasite, which is governed by $n_b$ and the off-rate $k_{\text{off}}$ (*Jana and Mognetti, 2019*). A significant increase of $n_b$ and/or a decrease of $k_{\text{off}}$ would lead eventually to parasite arrest (see section 'Effect of bond properties on parasite alignment'), which can be compared well with the model based on a homogeneous interaction potential (*Hillringhaus et al., 2019*).

There exist three different timescales which might be relevant for the parasite alignment: (i) bond lifetime $\tau_b \simeq 1/k_{\text{off}}$, (ii) membrane deformation time on the scale of parasite size $\tau_p \simeq \eta R_a^3/\kappa$, and (iii) rotational diffusion time of the parasite $\tau_r \simeq 8\pi\eta R_a^3/k_B T$. These characteristic times are $\tau_b \approx 0.013\,s$, $\tau_p \approx 0.011\,s$, and $\tau_r \approx 20\,s$ computed from the model parameters given in *Tables 1* and *2*. There is a clear separation of timescales between $\tau_r$ and both $\tau_b$ and $\tau_p$, indicating that the rotational diffusion of the parasite is too slow to have a significant effect on merozoite alignment. Furthermore, $\tau_b$ and $\tau_p$ are comparable in magnitude, suggesting that both bond dynamics and membrane deformations are important for the alignment process. It is also interesting to note that the ratio $\tau_p/\tau_r = k_B T/(8\pi\kappa) \approx 6 \times 10^{-4}$ depends only on the bending rigidity $\kappa$. This means that membrane deformation will always represent a dominating timescale over the rotational diffusion of the parasite, independently of the parasite size and the viscosity of suspending medium.

After the detailed analysis of parasite alignment, let us consider a possible influence of the effective membrane thickness, characterized by $\sigma$, on merozoite alignment. *Figure 5—figure supplement 2* presents various alignment characteristics for $\sigma = 0.15\,\mu m$ and $\sigma = 0.3\,\mu m$ in comparison with the original choice of $\sigma = 0.2\,\mu m$. The simulation results indicate that the $\sigma$ value may affect the number of bonds between the RBC and parasite, and thus the degree of membrane wrapping. This result is not entirely surprising, as $\sigma$ also affects the binding range defined as $2^{1/6}\sigma + \ell_{\text{eff}}^{\text{long}}$ and $2^{1/6}\sigma + \ell_{\text{eff}}^{\text{short}}$ for long and short ligands, respectively. However, differences in alignment results are rather small for $\sigma = 0.15\,\mu m$ and $\sigma = 0.2\,\mu m$, indicating that the choice for small enough $\sigma$ we made is appropriate. The case with $\sigma = 0.3\,\mu m$ exhibits a larger number of bonds and stronger membrane deformations than for $\sigma = 0.2\,\mu m$. Finally, note that fixed-time displacement characteristics of the parasite in *Figure 5—figure supplement 2* remain nearly unaffected by the $\sigma$ value, because dynamical properties of the merozoite are mainly determined by the bond off-rate, see the next section.

## Effect of bond properties on parasite alignment

To better understand the dependence of merozoite alignment on bond kinetics, the off-rate $k_{\text{off}}$ is varied for two ratios $k_{\text{on}}^{\text{short}}/k_{\text{on}}^{\text{long}}$ of short and long bond on-rates. *Figure 6* presents the parasite's fixed-time displacement, deformation energy, and average alignment times as a function of $k_{\text{off}}/k_{\text{on}}^{\text{long}}$. A lower ratio of $k_{\text{off}}/k_{\text{on}}^{\text{long}}$ (i.e. a lower $k_{\text{off}}$) leads to

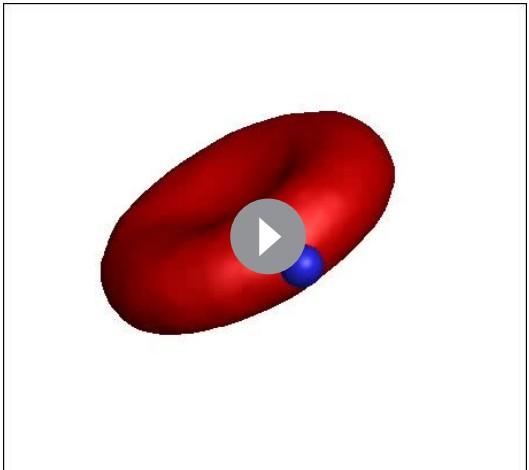

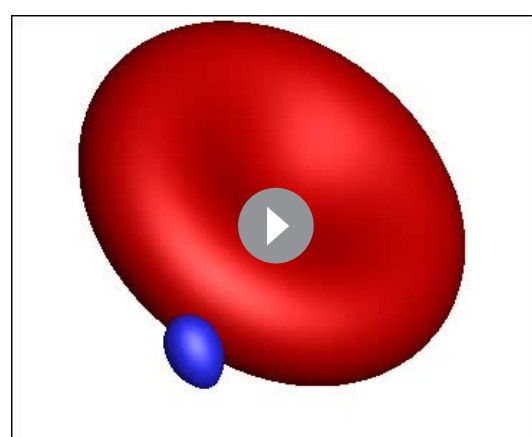

**Video 2.** Parasite adhesion and dynamics on a deformable RBC for a reduced off-rate $k_{\text{off}}$. $k_{\text{off}}/k_{\text{on}}^{\text{long}} = 1$.
https://elifesciences.org/articles/56500#video2

**Video 3.** Parasite dynamics at the surface of a rigid RBC for the reference RBC-parasite interactions from *Table 2*. $k_{\text{off}}/k_{\text{on}}^{\text{long}} = 2$.
https://elifesciences.org/articles/56500#video3

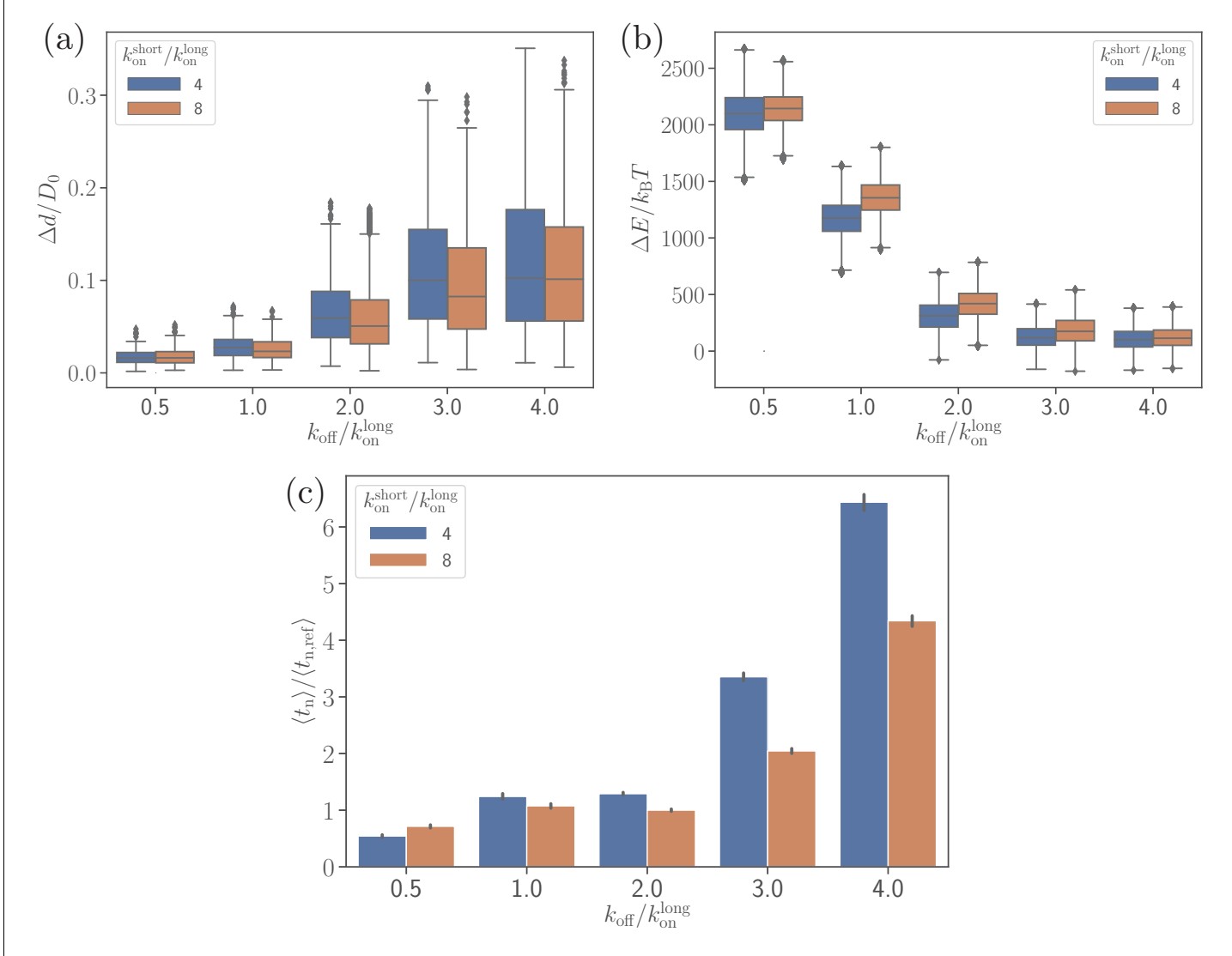

**Figure 6.** Effect of the off-rate $k_{\mathrm{off}}$ on (a) the parasite's fixed-time displacement, (b) RBC deformation energy, and (c) alignment time. Since the off-rate controls the lifetime of bonds, a smaller off-rate results in a stronger adhesion, a lower parasite displacement, and a faster alignment time.

The online version of this article includes the following source data and figure supplement(s) for figure 6:

**Source data 1.** Source data for graphs shown in *Figure 6a–c* and *Figure 6—figure supplement 1*.

**Figure supplement 1.** Effect of the off-rate $k_{\mathrm{off}}$ on (a) the apex distance, (b) alignment angle, and (c) the number of bonds.

stronger adhesion and thereby stronger membrane deformations (see *Figure 6b* and *Video 2*), consistently with the number of bonds shown in *Figure 6—figure supplement 1*. For small $k_{\mathrm{off}}/k_{\mathrm{on}}^{\mathrm{long}}$ values, membrane deformation energies can reach up to $2000\,\mathrm{k_B T}$, whereas large values of $k_{\mathrm{off}}$ result in $\Delta E_{\mathrm{rbc}} \approx 100\,\mathrm{k_B T}$. The main reason is that low values of $k_{\mathrm{off}}$ lead to a significant increase in the lifetime of individual bonds, allowing the parasite to form more bonds and thereby increase its adhesion energy and induce larger membrane deformations. Similarly, large values of $k_{\mathrm{off}}$ decrease the bond lifetime, resulting in a decrease in the adhesion energy. For instance, in case of $k_{\mathrm{off}}/k_{\mathrm{on}}^{\mathrm{long}} = 0.5$, the parasite forms on average about 200 bonds, whereas for $k_{\mathrm{off}}/k_{\mathrm{on}}^{\mathrm{long}} = 4$, the average number of bonds is approximately 15 (see *Figure 6—figure supplement 1*). Furthermore, a larger on-rate for the short bonds yields a slight increase in the strength of membrane deformations in comparison to a smaller $k_{\mathrm{on}}^{\mathrm{short}}$.

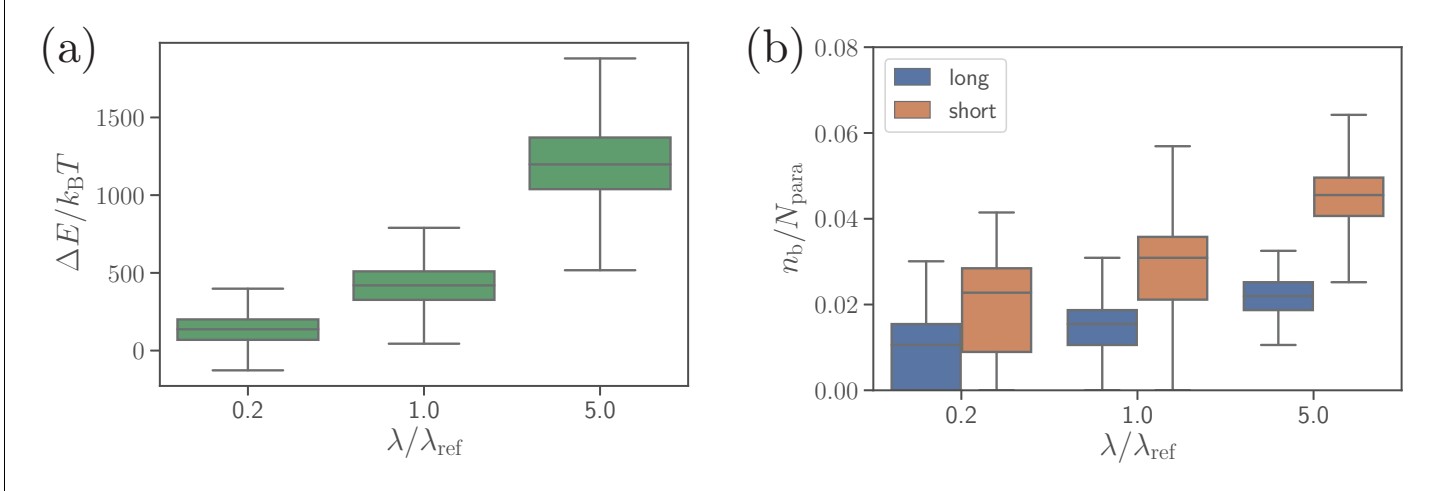

**Figure 7.** Effect of the extensional bond rigidities on parasite alignment. (a) RBC deformation energy and (b) the number of short and long bonds as a function of $\lambda/\lambda_{ref}$. $\lambda_{ref}$ corresponds to the reference case with parameters given in **Table 2**. Note that both $\lambda_{long}$ and $\lambda_{short}$ are changed by the same factor with respect to their $\lambda_{ref}$ values. Here, the bond kinetic rates are $k_{on}^{short} = 290.3\ \tau^{-1}$, $k_{on}^{long} = 36.3\ \tau^{-1}$, and $k_{off} = 72.6\ \tau^{-1}$.

The online version of this article includes the following source data and figure supplement(s) for figure 7:

**Source data 1.** Source data for graphs shown in **Figure 7a,b** and **Figure 7—figure supplement 1**.

**Figure supplement 1.** Effect of the extensional bond rigidities on (a) the apex distance, (b) alignment angle, and (c) fixed-time displacement of the parasite.

**Figure 6b,c** shows that there is a clear correlation between the level of membrane deformations and average alignment time. For example, for off-rates $k_{off}/k_{on}^{long} \leq 2$, the alignment times are comparable with those for the reference parameter case, while for off-rates $k_{off}/k_{on}^{long} > 2$, there is a strong increase in alignment times, which is correlated with insignificant membrane deformations. A shorter alignment time for $k_{off}/k_{on}^{long} \leq 2$ is due to the partial wrapping of the parasite by the RBC membrane, which is consistent with the previous study by *Hillringhaus et al., 2019* that demonstrates the importance of membrane deformation for merozoite alignment. Note that the fixed-time displacement $\Delta d$ in **Figure 6a** significantly increases with $k_{off}$ due to a weaker adhesion. This seems to imply that the parasite alignment may proceed faster for $k_{off}/k_{on}^{long} > 2$. However, as it is evident from **Figure 6c**, this simple expectation is not applicable here, indicating that a faster motion of the parasite at the RBC surface may not necessarily result in a faster alignment. Alignment times for $k_{on}^{short}/k_{on}^{long} = 8$ are generally shorter than for $k_{on}^{short}/k_{on}^{long} = 4$ because of a slightly stronger parasite wrapping by the membrane. A seemingly opposite result for $k_{off}/k_{on}^{long} = 0.5$ in **Figure 6c** is likely due to insufficient statistics in the probability maps used for MC sampling, as they are constructed based on several direct simulations. Accurate resolution of small differences in alignment times is challenging, as it requires a large number of direct simulations.

Another bond parameter, which may affect parasite alignment, is the extensional rigidities of both bond types. **Figure 7** presents RBC deformation energy and the number of bonds for five times softer and stiffer bonds than those in the reference case. Bonds with a larger rigidity lead to the formation of a larger number of bonds, more membrane wrapping, and a larger RBC deformation energy in comparison to soft bonds. The physical mechanism is that stiffer bonds facilitate a smaller distance between the membrane and the parasite at the edge of adhesion area between them, which favors further wrapping by the formation of additional bonds. Therefore, the spring rigidity in our model can mediate distance-limited bond formation at the edge of adhesion area between the parasite and the membrane, which affects merozoite alignment (see **Figure 7—figure supplement 1**), and is connected to membrane bending rigidity and the degree of wrapping. Consistently, simulations of the merozoite on a rigid RBC show no effect of the bond extensional rigidities on parasite alignment, because no significant membrane deformations are induced by parasite adhesion.

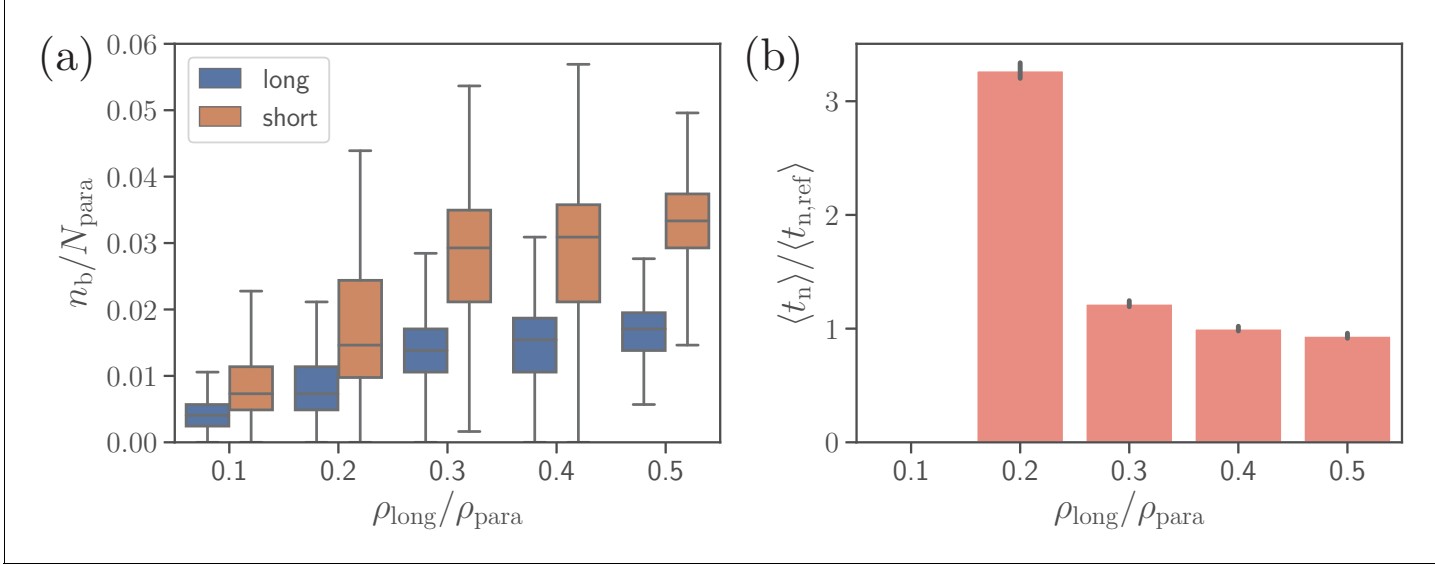

**Figure 8.** Effect of the density of long ligands $\rho_{\mathrm{long}}$ on parasite alignment. (a) Number of short and long bonds and (b) parasite alignment times as a function of $\rho_{\mathrm{long}}/\rho_{\mathrm{para}}$. Note that $\rho_{\mathrm{long}} + \rho_{\mathrm{short}} = \rho_{\mathrm{para}}$ remains constant in all simulations. Here, the bond kinetic rates are $k_{\mathrm{on}}^{\mathrm{short}} = 290.3\ \tau^{-1}$, $k_{\mathrm{on}}^{\mathrm{long}} = 36.3\ \tau^{-1}$, and $k_{\mathrm{off}} = 72.6\ \tau^{-1}$. In case of $\rho_{\mathrm{long}}/\rho_{\mathrm{para}} = 0.1$, parasite alignment time could not be computed through the MC sampling, since merozoite alignment has never occurred in direct simulations.

The online version of this article includes the following source data and figure supplement(s) for figure 8:

**Source data 1.** Source data for graphs shown in *Figure 8a,b* and *Figure 8—figure supplements 1* and *2*.

**Figure supplement 1.** Effect of the density of long ligands $\rho_{\mathrm{long}}$ on (a) deformation energy, (b) fixed-time displacement, (c) apex distance, and (d) alignment angle.

**Figure supplement 2.** Alignment results of simulations with only long ligands, i.e. for $\rho_{\mathrm{long}}/\rho_{\mathrm{para}} = 1$.

Furthermore, we consider effect of the density of long ligands $\rho_{\mathrm{long}}$ on parasite alignment. For the reference parameter set, $\rho_{\mathrm{long}}$ is chosen to be $\rho_{\mathrm{long}}/\rho_{\mathrm{para}} = 0.4$, so that $\rho_{\mathrm{short}}/\rho_{\mathrm{para}} = 0.6$. *Figure 8* presents the number of short and long bonds as well as parasite alignment times as a function of $\rho_{\mathrm{long}}/\rho_{\mathrm{para}}$. Interestingly, the number of short bonds increases with increasing $\rho_{\mathrm{long}}$, even though the density of short ligands $\rho_{\mathrm{short}}$ decreases. This occurs due to the fact that more long bonds further stabilize parasite adhesion, allowing the formation of more short bonds. Note that for the density $\rho_{\mathrm{long}}/\rho_{\mathrm{para}} = 0.1$ in *Figure 8b*, the value of $<t_n>$ is omitted, as the alignment criteria in *Equation 4* have not successfully been met during the entire course of direct simulations, yielding the probability of parasite alignment in MC sampling to be zero. For ligand densities $\rho_{\mathrm{long}}/\rho_{\mathrm{para}} \geq 0.3$, both bond numbers and alignment times remain nearly independent of $\rho_{\mathrm{long}}$. However, the average alignment time for $\rho_{\mathrm{long}}/\rho_{\mathrm{para}} = 0.2$ is about $30\,\mathrm{s}$ which is roughly three times longer than for the reference case. Note that $30\,\mathrm{s}$ is longer than the total length ($\approx 26\,\mathrm{s}$) of direct simulations. Nevertheless, parasite alignment has occurred in some of these simulations, resulting in a small non-zero probability of merozoite alignment and a relatively long $<t_n>$ calculated through the MC sampling. The fact that $<t_n>$ for $\rho_{\mathrm{long}}/\rho_{\mathrm{para}} = 0.2$ is longer than the total time of direct simulations means that the probability of parasite alignment is likely overestimated, indicating that the average alignment time should be even longer than $30\,\mathrm{s}$. An increase of $<t_n>$ with decreasing values of $\rho_{\mathrm{long}}$ is consistent with a significant decrease in membrane deformations (see *Figure 8—figure supplement 1*). For off-rates $k_{\mathrm{off}} < 72.6\ \tau^{-1}$, the trends illustrated in *Figure 8* remain qualitatively the same.

The importance of different ligand densities discussed above triggers the question whether both ligand types are necessary. Simulations performed with only short ligands (i.e., $\rho_{\mathrm{short}}/\rho_{\mathrm{para}} = 1$) for several different $k_{\mathrm{off}}$ rates show that the parasite is not able to achieve significant wrapping by the membrane, because such ligands are too short to facilitate progressive membrane attachment over a curved parasite surface. This limitation is directly connected to the density of available receptors on the RBC surface, which is determined in our model by the membrane resolution. For the same reason, parasite mobility is impaired, as it is largely mediated by bond formation/dissociation at the

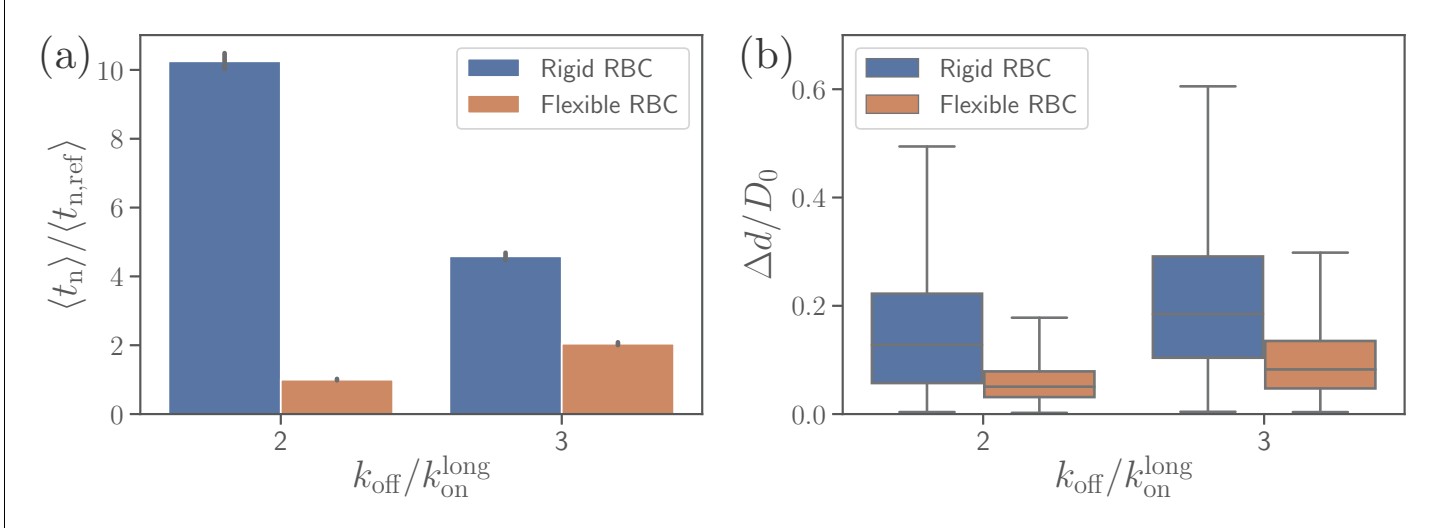

**Figure 9.** Effect of RBC membrane rigidity on (a) alignment time and (b) parasite fixed-time displacement for different off-rates $k_{off}$. Note that for a rigid RBC with $k_{off}/k_{on}^{long} = 1$, parasite alignment time could not be computed through the MC sampling, as the alignment criteria have never been met in direct simulations.

The online version of this article includes the following source data for figure 9:

**Source data 1.** Source data for graphs shown in *Figure 9a,b*.

edge of adhesion area between the parasite and the membrane. Therefore, the model with only short ligands does not reproduce proper parasite alignment. Simulations performed with only long ligands (i.e., $\rho_{long}/\rho_{para} = 1$) show that the parasite mobility and alignment can be well reproduced, see *Figure 8—figure supplement 2*. Thus, the presence of long bonds aids in the stabilization of merozoite adhesion and the enhancement of parasite motion, such that long bonds serve as some sort of effective leverages. Theoretically, a model with only long ligands would be sufficient to reproduce the proper parasite alignment; however, current biomolecular knowledge about parasite coating does not support the presence of many bonds with a length of about $100\,nm$. We speculate that short bonds are necessary (i) to stabilize parasite adhesion, as the density of long ligands is likely low, and (ii) to bring the two cells in sufficiently close contact (about $10\,nm$) to facilitate the formation of a tight junction required for invasion. Thus, the presence of both ligand types is likely necessary for a successful invasion.

## Effect of RBC rigidity

To investigate the effect of RBC rigidity on the alignment of a merozoite, we consider a nearly rigid cell membrane by increasing both bending rigidity and Young's modulus by two orders of magnitude in comparison to a healthy RBC. Such a rigid RBC shows no significant membrane deformations for the reference interaction parameters given in *Table 2*, see *Video 3*. Comparison of parasite fixed-time displacements and alignment times for flexible and rigid membranes is shown in *Figure 9* for two different values of $k_{off}$. Clearly, larger RBC rigidity leads to much longer parasite alignment times (see *Figure 9b*), emphasizing again the importance of membrane deformations for merozoite alignment. For off-rates $k_{off}/k_{on}^{long}<2$, parasite alignment at the surface of a rigid RBC is not achieved within the course of the simulation. As the off-rate increases, alignment time at the rigid membrane becomes comparable with that for the flexible membrane, because large enough $k_{off}$ values do not result in strong membrane deformations even for the flexible RBC. Thus, for large off-rates, the parasite's alignment solely relies on its rotational dynamics controlled by the bond kinetic rates.

*Figure 9a* presents a comparison of parasite fixed-time displacements at the flexible and rigid membranes. In both cases, parasite displacements increase with increasing $k_{off}$, as expected. However, the displacement at the rigid membrane is larger than at the flexible membrane (for visual comparison, see *Videos 1* and *3*), because the merozoite forms less bonds at the rigid surface. For the same reason, the variance of parasite displacements is larger for the rigid RBC than for the

flexible RBC. Note that an increase in $k_{\text{off}}$ results in an increase of fixed-time displacement and a decrease of alignment time for the rigid membrane, whereas for flexible RBC, an increase in off-rate leads to an elevation of both fixed-time displacement and alignment time. This implies that for a rigid RBC, fast kinetics or weak adhesion are favorable for a quick alignment. In contrast, for a flexible RBC, slow kinetics or strong adhesion are advantageous for fast alignment, since the parasite employs RBC deformation for efficient alignment by partial membrane wrapping.

## Discussion and conclusions

We have investigated the alignment of a merozoite at RBC membrane using a realistic two-state bond-dynamics model for parasite adhesion. Motivated by experiments (*Bannister et al., 1986*), parasite adhesion is modeled by two bond types, with long and short binding ranges. Since RBC-parasite interactions and the corresponding bond properties are experimentally not yet well characterized, the calibration of bond parameters is based on parasite fixed-time displacement at the membrane from existing experiments (*Weiss et al., 2015*), which is in the range of $0.3 - 0.8\,\mu m$. The presented model is able to reproduce quantitatively experimentally measured alignment times. Simulated alignment times are in the range between a few seconds and $26\,\text{s}$, while the analysis of experimental videos by *Weiss et al., 2015* yields an average alignment time of $16\,\text{s}$. Another independent experimental study by *Yahata et al., 2012* reports alignment times in the range between 7 and $44\,\text{s}$, which agree relatively well with our simulation predictions. In addition to the good agreement between simulated and experimental alignment times, our model reproduces well dynamic RBC membrane deformations frequently observed in experiments (*Dvorak et al., 1975*; *Gilson and Crabb, 2009*; *Crick et al., 2013*).

Our main result is that parasite alignment is mediated by RBC membrane deformations and a diffusive-like dynamics due to the stochastic nature of parasite-membrane interactions. Average number of bonds $<n_b>$ between the parasite and the membrane is governed by the ratio $k_{\text{on}}/k_{\text{off}} = \exp\left(\Delta U_b/k_{\text{B}}T\right)$ that is connected to the binding energy $\Delta U_b$ of a single bond and determines the strength of membrane deformations. Our results show that membrane deformations speed up the alignment through partial wrapping of the parasite, facilitating a contact between the parasite apex and the membrane. This conclusion is consistent with the previous simulation study (*Hillringhaus et al., 2019*), where merozoite adhesion has been modeled by a laterally homogeneous interaction potential whose strength controls RBC deformations. The importance of membrane deformation is also corroborated by simulations of parasite alignment at a rigid RBC, which show a drastic increase in alignment times. For a rigid membrane, the parasite alignment depends mainly on bond lifetime (i.e., $\tau_b \simeq 1/k_{\text{off}}$), indicating that a low $k_{\text{off}}$ or large bond lifetime may significantly decelerate the parasite's rotational motion, and hence, increase its alignment time drastically. This conclusion agrees well with a recent simulation study (*Jana and Mognetti, 2019*) on the dynamics of two adhered colloids, whose effective rotational diffusion is governed not only by $<n_b>$ but also by $\tau_b$. Clearly, $\tau_b$ is also important for parasite dynamics at a deformable RBC, in addition to the membrane relaxation time $\tau_p$ on the scale of parasite size. The poor alignment of the merozoite at a stiff membrane can be a contributing factor, limiting parasite invasion. For example, infected RBCs in malaria become significantly stiffer than healthy cells (*Suresh et al., 2005*; *Fedosov et al., 2011*), limiting secondary invasion events. Furthermore, an increased RBC membrane stiffness is relevant in many other diseases, such as sickle cell anemia (*Barabino et al., 2010*), thalassemia (*Peters et al., 2011*), and stomatocytosis (*Caulier et al., 2018*), whose carriers are generally less susceptible to malaria infection.

For large values of $k_{\text{off}}$, the parasite is not able to induce strong deformations even at a flexible membrane, so that the alignment times at rigid and deformable RBCs become comparable, and the alignment is governed solely by a diffusive-like rotational dynamics. The diffusive-like motion of the parasite at the membrane surface is facilitated by stochastic formation/dissociation of bonds between the two cell surfaces, and leads occasionally to a successful alignment. Therefore, our model is also able to explain the possibility of RBC invasion by a merozoite without preceding membrane deformations, which is observed much less frequently than the invasion preceded by significant RBC deformations (*Weiss et al., 2015*). Note that the RBC-parasite adhesion model based on a laterally homogeneous interaction potential (*Hillringhaus et al., 2019*) predicts the complete failure of parasite alignment without significant membrane deformations, because it does not capture a

diffusive-like rotational dynamics of the parasite. Thus, the bond-based model is more appropriate for the representation of RBC-parasite interactions.

Even though the bond parameters in *Table 2* were calibrated by the parasite fixed-time displacement obtained from experiments (*Weiss et al., 2015*), such a choice is likely not unique as some other set of parameters (e.g., receptor and ligand densities, bond rigidities and kinetic rates) may lead to statistically similar displacement characteristics. Nevertheless, it is important to emphasize that the discrete bonds in simulations should be thought of as 'effective' bonds, which likely represent a small cluster of real molecular bindings. Furthermore, since the parasite displacement is mainly controlled by the bond kinetics, this calibration procedure is rather robust in identifying an appropriate range of bond properties. Another important aspect of this model is the necessity of sufficiently long ligands and bonds to facilitate dynamic motion of the parasite at RBC surface. Simulations with only short ligands show that the parasite fails to induce significant wrapping by the membrane, leading to very little alignment success. Therefore, the long bonds serve as leverages for stable parasite adhesion and its motion at the membrane. Even though simulations with only long ligands indicate that a proper alignment can be achieved in this case, the existence of a dense population of long bonds has currently no support experimentally. Furthermore, we hypothesize that short enough bonds are necessary to enable the formation of a tight junction for parasite invasion, which requires a contact distance of about $10\,\text{nm}$ between the two cells. Thus, our simulations suggest that both ligand types are likely necessary.

Electron microscopy images of adhered parasites (*Bannister et al., 1986*) suggest that the density of long bonds can be as low as 5 - 10%. However, the density of long ligands and bonds in our simulations is limited by the resolution of both the RBC and parasite to be larger than about 20%. A much finer membrane model would alleviate this limitation, but it would be prohibitively expensive computationally. Note that such heterogeneous receptor-ligand interactions exist in other biological systems as well. For example, during leukocyte binding in the microvasculature, both selectin and integrin molecules participate in adhesion and work synergistically, even though they have distinct functions (*Ley et al., 2007*). Furthermore, infected RBCs in malaria adhere to endothelial cells via two distinct receptors, ICAM-1 and CD-36, where binding with ICAM-1 exhibits a catch-like bond, while the interaction with CD-36 is a slip-like bond (*Lim et al., 2017*).

Several studies (*Cowman et al., 2012*; *Dasgupta et al., 2014*; *Singh et al., 2010*) about RBC-parasite interactions hypothesize the existence of an adhesion gradient along the parasite body, which is expected to facilitate alignment. Based on the RBC-parasite adhesion model with a laterally homogeneous interaction potential (*Hillringhaus et al., 2019*), it was shown that an adhesion gradient, where the potential strength increases toward the apex of a merozoite, generally accelerates parasite alignment. No definite conclusions about possible gradients can be made in the context of that model, because even in the case of no adhesion gradients, it predicts very short alignment times of about two orders of magnitude smaller than measured experimentally. An introduction of adhesion gradients in our bond-based interaction model leads qualitatively to the following conclusions: (i) Weak adhesion gradients do not significantly disturb the irregular motion of a parasite at RBC membrane, and have a negligible effect on the alignment. (ii) Strong adhesion gradients often result in a controlled direct re-orientation of the parasite toward its apex, suppressing the irregular motion observed experimentally. These preliminary results do not permit a definite conclusion about the possible existence of adhesion gradients, as moderate adhesion gradients may exist and aid partially in the alignment process. Nevertheless, our model shows that adhesion gradients are not necessary, since the main parasite properties, such as dynamic motion and realistic alignment times, can be reproduced well by the bond-based model without adhesion gradients.

In conclusion, our model suggests that the parasite alignment can be explained by the passive compliance hypothesis (*Introini et al., 2018*; *Hillringhaus et al., 2019*), such that no additional active mechanisms or processes are necessary. Of course, this does not eliminate the possible existence of some active mechanisms, which may participate in the alignment process. Another limitation of many studies is that the parasite alignment is investigated under static (no flow) conditions, whereas in vivo, parasite alignment and invasion occur under a variety of blood flow conditions, including different flow stresses and flow-induced RBC deformations (*Lanotte et al., 2016*). Further experiments are needed to investigate RBC-parasite interactions for realistic blood-flow scenarios. The bond-based model proposed here is expected to be useful for the quantification of such

experimental studies and for a better understanding of RBC-parasite adhesion under blood flow conditions.

## Model and methods

### Red blood cell model

The total potential energy of the RBC model is given by *Fedosov et al., 2010a*; *Fedosov et al., 2010b*

$$U_{\text{rbc}} = U_{\text{sp}} + U_{\text{bend}} + U_{\text{area}} + U_{\text{vol}}. \tag{6}$$

Here, the term $U_{\text{sp}}$ represents the elasticity of spectrin network, which is attached to the back side of the lipid membrane. $U_{\text{bend}}$ models the resistance of the lipid bilayer to bending. $U_{\text{area}}$ and $U_{\text{vol}}$ constrain the area and volume of RBC membrane, mimicking incompressibility of the lipid bilayer and the cytosol, respectively.

The elastic energy term $U_{\text{sp}}$ is given by

$$U_{\text{sp}} = \sum_{i=1}^{N_s} \frac{k_{\text{B}} T \ell_i^{\text{max}} \left(3x_i^2 - 2x_i^3\right)}{4p_{\text{i}}(1-x_i)} + \frac{\lambda_i}{\ell_i}, \tag{7}$$

where the first term is the attractive worm-like chain potential, while the second term corresponds to a repulsive potential with a strength $\lambda_i$. Furthermore, $\ell_i$ is the length of the i-th spring, $p_i$ is the persistence length, $\ell_i^{\text{max}}$ is the maximum extension, and $x_i = \ell_i/\ell_i^{\text{max}}$. The stress-free state of the elastic network is considered to be a biconcave RBC shape, such that initial lengths in the triangulation of this shape define equilibrium spring lengths $\ell_i^0$. For a regular hexagonal network, its two-dimensional (2D) shear modulus $\mu$ can be derived in terms of model parameters as (*Fedosov et al., 2010a*; *Fedosov et al., 2010b*)

$$\mu = \frac{\sqrt{3}k_{\text{B}} T}{4p_i\ell_i^0} \left( \frac{\bar{x}}{2(1-\bar{x})^3} - \frac{1}{4(1-\bar{x})^2} + \frac{1}{4} \right) + \frac{3\sqrt{3}\lambda_i}{4(\ell_i^0)^3}, \tag{8}$$

where $\bar{x} = \ell_i^0/\ell_i^{\text{max}}$ is a constant for all $i$. Thus, for given values of $\mu$, $\bar{x}$, and $\ell_i^0$, individual spring parameters $p_i$ and $\lambda_i$ are calculated by using *Equation 8* and the force balance $\partial U_{\text{sp}}/\partial l_i|_{\ell_i^0} = 0$ for each spring.

The bending energy of the membrane is expressed as (*Gompper and Kroll, 1996*; *Gompper and Kroll, 2004*)

$$U_{\text{bend}} = \frac{\kappa}{2} \sum_{i=1}^{N_{\text{rbc}}} \frac{1}{\sigma_i} \left[ \mathbf{n}_i^{\text{rbc}} \cdot \left( \sum_{j(i)} \frac{\sigma_{ij}}{r_{ij}} \mathbf{r}_{ij} \right) \right]^2 \tag{9}$$

where $\kappa$ is the bending modulus, $\mathbf{n}_i^{\text{rbc}}$ is a unit normal of the membrane at vertex $i$, $\sigma_i = \left( \sum_{j(i)} \sigma_{ij} r_{ij} \right)/4$ is the area of dual cell of vertex $i$, and $\sigma_{ij} = r_{ij}[\cot(\theta_1) + \cot(\theta_2)]/2$ is the length of the bond in dual lattice, with the two angles $\theta_1$ and $\theta_2$ opposite to the shared bond $\mathbf{r}_{ij}$.

The last two terms in *Equation 6*,

$$U_{\text{area}} = \frac{k_{\text{a}}(A-A_0)^2}{2A_0} + \sum_{i=1}^{N_{\text{t}}} \frac{k_\ell \left(A_i - A_i^0\right)^2}{2A_i^0},$$

$$U_{\text{vol}} = \frac{k_{\text{v}}(V-V_0)^2}{2V_0}, \tag{10}$$

constrain surface area and volume of the RBC (*Fedosov et al., 2010a*; *Fedosov et al., 2010b*), where $k_{\text{a}}$ and $k_\ell$ control the total surface area $A$ and local areas $A_i$ of each triangle to be close to desired total area $A_0$ and local areas $A_i^0$, respectively. The coefficient $k_{\text{v}}$ controls the total volume $V$ of the cell. The values of these coefficients are chosen large enough such that the area and volume fluctuate within 1% of the desired values.

The elasticity of a healthy RBC is characterized by the 2D shear modulus $\mu \approx 4.8 \mu \mathrm{Nm}^{-1}$, which corresponds to the 2D Young's modulus $Y \approx 18.9 \mu \mathrm{Nm}^{-1}$ for a nearly incompressible membrane (*Suresh et al., 2005*; *Fedosov et al., 2010a*). These values are employed in all simulations unless stated otherwise. The described membrane model has been shown to accurately capture RBC mechanics (*Fedosov et al., 2010a*; *Fedosov et al., 2010b*) and membrane fluctuations (*Turlier et al., 2016*).

## RBC-parasite adhesion interaction

Interaction between parasite and RBC membrane has two components. The first part imposes excluded-volume interactions between the RBC and merozoite (i.e. no overlap between them), using the purely repulsive part of the Lennard-Jones (LJ) potential

$$U_{\mathrm{rep}}(r) = 4\epsilon \left[ \left( \frac{\sigma}{r} \right)^{12} - \left( \frac{\sigma}{r} \right)^{6} \right], \quad r \leq 2^{1/6}\sigma. \tag{11}$$

This potential acts between every pair of RBC and parasite vertices separated by a distance $r = |\mathbf{r}_{\mathrm{rbc}} - \mathbf{r}_{\mathrm{para}}|$ that is smaller than $2^{1/6}\sigma$. Here, $\epsilon$ represents the strength of interaction and $\sigma$ is the characteristic length scale of repulsion. The distance $\sigma$ can be thought of as an effective membrane thickness (imagine a surface constructed from overlapping spheres with a diameter $\sigma$). Normally, $\sigma$ should be selected as small as possible for a given resolution length of both the RBC membrane and parasite, which is about 0.2 μm in our models. Therefore, $\sigma = 0.2\,\mu m$ is chosen, such that no overlap between the cells is guaranteed and the interacting surface is smooth enough.

The attractive part of RBC-parasite interaction is modeled by a reversible two-state bond model. Bonds can form between RBC membrane vertices representing receptors and merozoite vertices corresponding to ligands, while existing bonds can also dissociate. These bonds represent RBC-parasite adhesion through existing agonists at the surface of these cells and can be formed by two different types of ligands:

i. long ligands with an effective binding range $\ell_{\mathrm{eff}}^{long} = 100\,nm$,
ii. short ligands with an effective binding range $\ell_{\mathrm{eff}}^{short} = 20\,nm$,

which is motivated by electron microscopy observations of RBC-merozoite adhesion (*Bannister et al., 1986*). Long ligands result in long bonds, while short ligands lead to short bonds. Both bond types are modeled by harmonic springs with the potential energy given by

$$U_{\mathrm{ad}}(\ell) = \frac{\lambda_{\mathrm{type}}}{2}(\ell - \ell_0)^2, \tag{12}$$

where $\lambda_{\mathrm{type}}$ is the spring extensional rigidity of either long or short bond type and $\ell_0$ is the equilibrium bond length. To model the dynamic two-state interaction, *constant* (i.e. length independent) on- and off-rates ($k_{\mathrm{on}}^{\mathrm{short}}$, $k_{\mathrm{on}}^{\mathrm{long}}$, and $k_{\mathrm{off}}$) are chosen, in order to simplify the model and reduce the number of parameters. Furthermore, the off-rate for both bond types is selected to be same. Note that this model can easily be extended to length-dependent rates.

To implement the different bond types, each vertex at the parasite surface represents either a long or a short ligand. The choice of vertices that correspond to long or short ligands is made randomly for fixed ligand densities $\rho_{\mathrm{long}}$ and $\rho_{\mathrm{short}}$. To avoid possible artifacts of a single discrete ligand distribution, each independent simulation assumes a different random choice of ligands with their respective densities kept constant. Bonds between the vertices at the RBC and parasite surfaces can form if the distance between two vertices is smaller than the corresponding cut-off distances $\ell_0 + \ell_{\mathrm{eff}}^{\mathrm{long}}$ and $\ell_0 + \ell_{\mathrm{eff}}^{\mathrm{short}}$, which remain the same in all simulations. Here, $\ell_0 = 2^{1/6}\sigma$ corresponds to the length of the excluded-volume LJ interactions between the vertices of RBC and parasite, whose choice is defined by a characteristic discretization length of the RBC membrane. Only a single bond is allowed at each vertex for the both ligand types. Note that existing bonds can stretch beyond their effective binding ranges $\ell_{\mathrm{eff}}^{\mathrm{long}}$ and $\ell_{\mathrm{eff}}^{\mathrm{short}}$.

## Hydrodynamic interactions

Hydrodynamic interactions are modeled using the dissipative particle dynamics (DPD) method (*Hoogerbrugge and Koelman, 1992*; *Español and Warren, 1995*), where fluid is represented by a collection of particles interacting through three types of pairwise forces: conservative $\mathbf{F}_{ij}^C$, dissipative $\mathbf{F}_{ij}^D$, and random $\mathbf{F}_{ij}^R$ forces. The total force between particles $i$ and $j$ is given by

$$\mathbf{F}_{ij} = \mathbf{F}_{ij}^C + \mathbf{F}_{ij}^D + \mathbf{F}_{ij}^R. \tag{13}$$

The conservative force models fluid compressibility, whereas the dissipative and random forces maintain a desired temperature of the system. The dissipative force also gives rise to fluid viscosity, which is generally measured in DPD by simulating a reversible-Poiseuille flow (*Backer et al., 2005*; *Fedosov et al., 2010c*). The DPD interactions are implemented only between the pairs of fluid-fluid, fluid-RBC, and fluid-parasite particles. DPD interaction parameters are selected such that they impose no-slip boundary condition at RBC and parasite surfaces (*Fedosov et al., 2010a*; *Hillringhaus et al., 2019*).

## Simulation setup

Simulation domain with a size of $7.7D_0 \times 3.1D_0 \times 3.1D_0$ contains both RBC and parasite suspended in a DPD fluid, where $D_0 = \sqrt{A_0/\pi}$ is the effective RBC diameter. Periodic boundary conditions are imposed in all directions. Initially, the parasite is placed close enough to the RBC membrane, so that the interaction between them is immediately possible. The initial parasite orientation is with its apex directed away from the membrane to mimic least favorable attachment configuration.

The main simulation parameters are shown in *Table 1*, both in simulation and physical units. To compare simulation units to physical units, a basic length scale is defined as the effective RBC diameter $D_0$, an energy scale as $k_\mathrm{B}T$, and a time scale as RBC membrane relaxation time $\tau = \eta D_0^3/\kappa$. For average properties of a healthy RBC, the effective diameter is $\mathrm{D}_0 \simeq 6.5\mu\mathrm{m}$ with $\mathrm{A}_0 = 133.5\mu\mathrm{m}^2$ (*Evans and Skalak, 1980*) and the relaxation time becomes $\tau \approx 0.92\,\mathrm{s}$ for the bending modulus $\kappa = 3 \times 10^{-19}\,J$ (*Evans, 1983*; *Fedosov et al., 2010a*) and plasma viscosity $\eta = 1\,\mathrm{mPa\,s}$. All simulations are performed on the supercomputer JURECA *Jülich Supercomputing Centre, 2018* at the Jülich Supercomputing Centre, Forschungszentrum Jülich.

## Monte-Carlo sampling of alignment times

One of the main foci of our study is to obtain distributions of parasite alignment times for various conditions, which requires a large number of simulations of merozoite alignment. In order to significantly reduce the computational effort, Monte-Carlo (MC) sampling of alignment times, which is guided by direct DPD simulations of RBC-parasite adhesion, is employed. The MC sampling is based on a two-dimensional probability map (see e.g. *Figure 4a*), which characterizes parasite orientation at the membrane surface through the distance $d_\mathrm{apex}$ between the parasite apex and membrane and merozoite alignment angle $\theta$ (see *Figure 3a* for definitions of $d_\mathrm{apex}$ and $\theta$). To construct such a probability map, possible $d_\mathrm{apex}$ and $\theta$ values are binned into a number of orientation states $(i,j) = (d_\mathrm{apex}^i, \theta^j)$, and the probability $P(i,j)$ of each state is computed from at least 10 long DPD simulations of RBC-parasite adhesion. We have verified that 10 independent DPD simulations are enough to reliably compute a probability map through its convergence with the number of DPD simulations. In the MC algorithm, changes in parasite orientation are modeled by transitions between different states, using the Metropolis algorithm. Thus, the transition from a state $(i,j)$ to one of the neighboring states $(i + 1,j)$, $(i - 1,j)$, $(i,j + 1)$ or $(i,j - 1)$ is selected randomly with a probability of $1/4$, and this move is accepted if $\xi < P(\mathrm{new\,state})/P(i,j)$, where $\xi$ is a random number drawn from a uniform distribution in the interval $[0, 1]$. In summary, the MC sampling algorithm is performed as follows:

1. Initial parasite orientation is selected randomly by choosing a state $(d_\mathrm{apex}^i, \theta^j)$, which has a non-zero probability.
2. Transitions between the neighboring states are modeled according to the Metropolis algorithm described above.
3. MC procedure is stopped whenever pre-defined alignment criteria are reached, and the number of MC steps is interpreted as alignment time.

Note that the MC sampling algorithm fulfills detailed balance, but does not account for hydrodynamic interactions. The fulfillment of detailed balance for the Metropolis algorithm in equilibrium means that changes between different states $(i,j)$ and $(i',j')$ (with energies $E_{(i,j)}$ and $E_{(i',j')}$, respectively) are performed according to transition rates proportional to $\exp\left[-(E_{(i,j)} - E_{(i',j')})/(k_B T)\right]$, which are directly connected to probabilities of different states. Noteworthy, the MC sampling is a fast and efficient way to sample the distribution of parasite alignment times.

## Acknowledgements

We would like to express our gratitude to Virgilio L Lew and Pietro Cicuta from the University of Cambridge for insightful discussions. Sebastian Hillringhaus acknowledges support by the International Helmholtz Research School of Biophysics and Soft Matter (IHRS BioSoft). We gratefully acknowledge the computing time granted through JARA-HPC on the supercomputer JURECA (*Jülich Supercomputing Centre, 2018*) at Forschungszentrum Jülich.

## Additional information

### Funding

The authors declare that there was no funding for this work.

### Author contributions

Sebastian Hillringhaus, Software, Formal analysis, Investigation, Visualization, Methodology, Writing - original draft; Anil K Dasanna, Formal analysis, Investigation, Visualization, Methodology, Writing - original draft; Gerhard Gompper, Conceptualization, Project administration, Writing - review and editing; Dmitry A Fedosov, Conceptualization, Software, Supervision, Project administration, Writing - review and editing

### Author ORCIDs

Sebastian Hillringhaus ⒾⒹ https://orcid.org/0000-0003-0100-9368
Anil K Dasanna ⒾⒹ https://orcid.org/0000-0001-5960-4579
Gerhard Gompper ⒾⒹ https://orcid.org/0000-0002-8904-0986
Dmitry A Fedosov ⒾⒹ https://orcid.org/0000-0001-7469-9844

### Decision letter and Author response
Decision letter https://doi.org/10.7554/eLife.56500.sa1
Author response https://doi.org/10.7554/eLife.56500.sa2

## Additional files

### Supplementary files
• Transparent reporting form

### Data availability
All data generated or analysed during this study are included in the manuscript and supporting files. Source data files have been provided for Figures 2-9, including all figure supplements. Figure 1 is a model schematic, which does not contain any data.

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
