## [Decision Letter]

**Acceptance summary:**

The paper by Hillringhaus et al. studies the invasion of red blood cells by malaria parasites (merozoites), a key element of their reproduction cycle during the blood stage of the disease. Building on earlier work demonstrating the importance of geometrical alignment of merozoites with the cell, adhesion to the cell membrane and binding by filaments, the present work develops a computational model that incorporates stochastic deformations of the cell membrane and the discrete nature of the adhesive bonds. By exploring the influence of various parameters, such as the bond kinetics and RBC membrane stiffness, it is shown that alignment times similar to those observed experimentally can be obtained purely by a passive mechanism that balances adhesion and membrane elasticity.

**Decision letter after peer review:**

Thank you for submitting your article "Stochastic bond dynamics facilitates alignment of malaria parasite at erythrocyte membrane upon invasion" for consideration by *eLife*. Your article has been reviewed by two peer reviewers, and the evaluation has been overseen by a Reviewing Editor and Suzanne Pfeffer as the Senior Editor The following individuals involved in review of your submission have agreed to reveal their identity: Michael Gomez (Reviewer #2).

The reviewers have discussed the reviews with one another and the Reviewing Editor has drafted this decision to help you prepare a revised submission.

This manuscript studies the invasion of red blood cells (RBCs) by malaria parasites (merozoites), a key element of their reproduction cycle during the blood stage of the disease. While initial attachment of merozoites to RBCs occurs rapidly, the major factor that limits successful invasion is that the merozoite must align almost perpendicularly at its apex to the RBC membrane. Using static models, previous work (e.g. Dusgupta et al., 2014 and Hillringhaus et al., 2019) has demonstrated the importance of adhesion (arising from filaments on the merozoite surface that bind to the RBC membrane) and elasticity of the RBC membrane: together these enable partial wrapping of the membrane around the merozoite to aid alignment. This manuscript builds upon these studies to address the dynamics of alignment, developing a computational model that incorporates stochastic deformations of the RBC membrane and the discrete nature of the adhesive bonds. By exploring the influence of various parameters, such as the bond kinetics and RBC membrane stiffness, they demonstrate that alignment times similar to those observed experimentally can be obtained purely by a passive mechanism, namely the balance between adhesion and membrane elasticity.

The manuscript is very well written and discusses the details of the model and its results thoroughly and clearly. The inclusion of dynamic effects and discrete bonds resolves major shortfalls of previous models. The resolution of the simulations is very impressive and enables new insight into what governs the timescale of alignment and hence the invasion process. In addition, it is clear that the framework developed here can be adapted and built upon in future work to explore other effects, as well as informing further experimental studies in this area. I therefore strongly recommend publication.

The reviewers request that you address the following issues to enhance the present story.

1) A numerical parameter that could potentially strongly influence the results is the repulsion distance *σ*. It must be very carefully checked how a variation in *σ* affects (or not) the results. This is particularly important since the authors do not only aim at a qualitative explanation, but a fully quantitative prediction of the biophysical alignment process.

2) The authors very briefly state in the Discussion that “simulations with only short bonds show that the parasite is quickly arrested…". The authors might include some data on that. Also, it triggers the question what happens if there are only long bonds? Or, to state the question somewhat deeper: is the two-bond combination really necessary to reproduce the alignment? Or can one imagine that a single bond, of whatever nature, reproduces the alignment equally well?

3) Again, in the Discussion the authors mention the “stochastic motion observed experimentally”. Figure 2B only shows the average fixed-time displacement, it does not indicate whether that motion is directed or truly stochastic. The authors should find a way to substantiate their claim that the experimental motion is truly stochastic and not somehow directed. For example, one might try to identify the (signed) distribution of Δ d within some meaningful local coordinates and see if the average is 0. Other ways to demonstrate the stochastic nature are certainly possible as well.

4) The effect of bond kinetics on alignment is discussed in detail, but what about the influence of the bond spring stiffnesses, i.e. λ_long_ and λ_short_ (defined in Equation 12)? I would guess that for a given bond number, the ratio of these stiffnesses to the membrane bending stiffness controls the degree of wrapping, similar to the dimensionless adhesion strength defined by Dusgupta et al., 2014. I appreciate much remains unknown about the properties of the binding filaments, but at the very least the values of λ_long_ and λ_short_ chosen in the model (Table 2) need some discussion (for example are they varied as part of the fitting procedure discussed in subsection “Calibration of RBC-parasite interactions”?).

5) Similarly, it would be good to know what influences the most-likely values of d_apex_ and *θ* for the distributions in Figures 3A-B. I understand that these peaks correspond to a configuration with maximum contact area (as discussed in subsection “Membrane deformation and parasite dynamics”). Are the associated values of *d_apex_* and *θ* then just determined by the egg-like geometry of the parasite and RBC, or do they also depend on the mechanical properties? Since these most-likely values (and their closeness to the values needed for alignment) play a key role in determining whether alignment occurs, it would be good to discuss what parameters control these values. I appreciate a thorough exploration of parameter space is not possible due to the computational time, but it would be worth at least having a qualitative idea.

6) Did the authors examine where on the RBC membrane alignment is most likely to occur? I would expect it to be the region near the centre of the RBC where the membrane is concave-outward, since here the membrane would naturally curve towards the merozoite so that wrapping requires less bending energy. This concave-outward region only exists on one side of sickle-shaped RBCs, which could be a contributing factor (as well as the increased membrane stiffness) as to why sickle cell anemia gives some resistance to malaria.

7) Similarly, is it possible to speculate on the influence of the merozoite shape on achieving alignment? For example, as well as providing an obvious advantage for initiating invasion, the egg shape means that the apex is titled towards the membrane in the most-likely configuration with maximum contact area (i.e. the peak of the distribution in Figure 3B is well above *θ* = π/2). This is not the case for a spherical shape, for which there is no preferred orientation, so that the merozoite may become more easily arrested with its apex pointing away from the RBC. Moreover, the tapering near the apex means there is naturally less material in its vicinity, so that alignment potentially could be achieved with an apex angle *θ* further from π.

8) Figure 4B: Why are there so many instances of very short alignment times, below the lower bound of 7 seconds observed experimentally by Yahata et al., 2012? Is this due to an effect being neglected or overestimated, or can the discrepancy be improved simply by tightening the alignment criteria (e.g. requiring *θ* > 0.9π rather than 0.8π)?

9) One concern is the sensitivity of the results to the discretisation used. This is because alignment requires that the distance between the merozoite apex and RBC membrane is very small and may become comparable to the discretisation length of the RBC membrane. The alignment criteria also necessitate examining small changes in the apex angle *θ* from π. For example, the change in the angle of the normal vector to the RBC membrane from one triangular face to the next scales as l/R, where l is the characteristic size of the triangles and R is the typical radius of curvature of the membrane. For the change in angle to be small (so that discretisation effects are negligible) requires l << R. However, the schematic in Figure 3A suggests that if the discretisation length is halved so that each segment becomes two segments, the value of *θ* could easily change by an amount comparable to the tolerance in the alignment criterion, i.e. 0.2π. Could the authors comment on this?

---

## [Author Response]

1) A numerical parameter that could potentially strongly influence the results is the repulsion distance σ. It must be very carefully checked how a variation in σ affects (or not) the results. This is particularly important since the authors do not only aim at a qualitative explanation, but a fully quantitative prediction of the biophysical alignment process.

First, let us point out that the repulsion interaction is required to prevent the overlap between discrete models of the parasite and membrane. The repulsive distance *σ* can be thought of as an effective membrane thickness (imagine a surface constructed from overlapping spheres with a diameter *σ*), to which additional interaction range l_eff_ is added. Normally, *σ* should be selected as small as possible for a given resolution length of both the RBC membrane and parasite (about 0.2 µm in our models). We selected *σ*=0.2 µm such that no overlap between the cells is guaranteed and the interacting surface is smooth enough. Thus, a little effect of the exact *σ* value on the adhesion results is expected.

In order to test this expectation, we have performed two new sets of simulations for a smaller (*σ*=0.15 µm) and larger (*σ*=0.3 µm) repulsion distances than *σ*=0.2 µm used previously. The results indicate that *σ* can have an effect on the number of bonds between the RBC and parasite as it affects the binding range defined as 2^1/6^*σ* + l_eff_. Differences in results (including deformation energy, number of bonds, and fixed-time parasite displacement) are rather small for *σ*=0.15 µm and *σ*=0.2 µm. The case with *σ*=0.3 µm exhibits a larger number of bonds than for *σ*=0.2 µm, resulting in stronger membrane deformations. Nevertheless, fixed-time displacement characteristics of the parasite remain nearly unaffected by *σ*. We have added these new results related to different *σ* values to the manuscript.

2) The authors very briefly state in the Discussion that “simulations with only short bonds show that the parasite is quickly arrested…". The authors might include some data on that. Also, it triggers the question what happens if there are only long bonds? Or, to state the question somewhat deeper: is the two-bond combination really necessary to reproduce the alignment? Or can one imagine that a single bond, of whatever nature, reproduces the alignment equally well?

This is a very interesting and important point. The previously included brief statement about short bonds was based on a single simulation, providing an inaccurate description of the complexity of the system. We have performed a number of additional simulations with only short and long ligands separately. Simulations with only short ligands for several different *k*_off_ rates show that the parasite is not able to achieve significant wrapping by the membrane, because such ligands are too short to facilitate progressive membrane attachment over a curved parasite surface. This limitation is directly connected to the density of available receptors on the RBC surface, which is determined in our model by the membrane resolution. For the same reason, parasite mobility is impaired, as it is largely mediated by bond formation/dissociation at the edge of adhesion area between the parasite and the membrane. Therefore, the model with only short ligands does not reproduce proper parasite alignment.

Simulations with only long ligands show that the parasite mobility and alignment can be well reproduced. Therefore, the presence of long bonds aids in the stabilization of parasite adhesion and the enhancement of parasite motion, such that long bonds serve as some sort of effective leverages. Theoretically, a model with only long ligands would be sufficient to reproduce the proper parasite alignment; however, current biomolecular knowledge about parasite coating does not support the presence of many bonds with a length of about 100 nm. We speculate that short bonds are necessary (i) to stabilize parasite adhesion, as the density of long ligands is likely low, and (ii) to bring the two cells in sufficiently close contact to facilitate the formation of a tight junction required for invasion. Thus, the presence of both ligand types is likely necessary for a successful invasion.

We have incorporated these new results and their Discussion into the manuscript.

3) Again, in the Discussion the authors mention the “stochastic motion observed experimentally”. Figure 2B only shows the average fixed-time displacement, it does not indicate whether that motion is directed or truly stochastic. The authors should find a way to substantiate their claim that the experimental motion is truly stochastic and not somehow directed. For example, one might try to identify the (signed) distribution of Δ d within some meaningful local coordinates and see if the average is 0. Other ways to demonstrate the stochastic nature are certainly possible as well.

Unfortunately, the amount of data (only 20 points) for the fixed-time displacement from experiments does not allow us to assess whether the motion is truly stochastic or not. Therefore, we modified the text stating that the parasite motion in experiments visually resembles an irregular motion, but it may also be partially directed. In simulations, the motion is stochastic and determined by the probability map (not a uniform distribution) in Figure 4A.

4) The effect of bond kinetics on alignment is discussed in detail, but what about the influence of the bond spring stiffnesses, i.e. λ_long_ and λ_short_ (defined in Equation 12)? I would guess that for a given bond number, the ratio of these stiffnesses to the membrane bending stiffness controls the degree of wrapping, similar to the dimensionless adhesion strength defined by Dusgupta et al., 2014. I appreciate much remains unknown about the properties of the binding filaments, but at the very least the values of λ_long_ and λ_short_ chosen in the model (Table 2) need some discussion (for example are they varied as part of the fitting procedure discussed in subsection “Calibration of RBC-parasite interactions”?).

We thank the reviewers for making this point. Our original expectation was that bond stiffnesses play a secondary role, because we employ constant (force independent) bond rates. To clarify this point, we have performed simulations for softer and stiffer bonds, which confirm that the reviewer’s expectations are qualitatively correct. Bonds with a larger stiffness lead to more membrane wrapping in comparison to soft bonds. The physical mechanism is that stiffer bonds facilitate a smaller distance between the membrane and the parasite at the edge of adhesion area between them, which favors further wrapping by the formation of additional bonds. Therefore, the spring stiffness in our model can mediate distance-limited bond formation at the edge of adhesion area between the parasite and the membrane, which is connected to membrane bending rigidity and the degree of wrapping. We have added these results into the revised manuscript with a short discussion.

5) Similarly, it would be good to know what influences the most-likely values of d_apex_ and θ for the distributions in Figures 3A-B. I understand that these peaks correspond to a configuration with maximum contact area (as discussed in subsection “Membrane deformation and parasite dynamics”). Are the associated values of d_apex_ and θ then just determined by the egg-like geometry of the parasite and RBC, or do they also depend on the mechanical properties? Since these most-likely values (and their closeness to the values needed for alignment) play a key role in determining whether alignment occurs, it would be good to discuss what parameters control these values. I appreciate a thorough exploration of parameter space is not possible due to the computational time, but it would be worth at least having a qualitative idea.

To check whether the most likely values of *d_apex_* and *θ* are mainly determined by the egg-like shape, or also depend on the mechanical properties of the membrane, we have computed *d_apex_* and θ distributions for a parasite adhered to membranes of various rigidities. Clearly, for the case of a rigid membrane, the most likely *d_apex_* and *θ* values are determined by the parasite shape corresponding to a configuration with maximum contact area. In comparison to the soft membrane, the peak in *d_apex_* for the rigid RBC is shifted further away from zero. This indicates that the degree of wrapping also affects the most likely values of *d_apex_* and *θ*. Therefore, in addition to the egg-like shape of a parasite, RBC membrane properties, such as bending rigidity, shear elasticity, and local curvature (see also our response to the next question), do affect the most likely values of *d_apex_* and *θ*.

We discuss now this issue in the revised manuscript in more detail.

6) Did the authors examine where on the RBC membrane alignment is most likely to occur? I would expect it to be the region near the centre of the RBC where the membrane is concave-outward, since here the membrane would naturally curve towards the merozoite so that wrapping requires less bending energy. This concave-outward region only exists on one side of sickle-shaped RBCs, which could be a contributing factor (as well as the increased membrane stiffness) as to why sickle cell anemia gives some resistance to malaria.

The RBC-membrane location of possible preferred adhesion is an important issue. We have re-checked the simulation data for the most likely alignment spot, and indeed the alignment occurs more often in one of the RBC dimples. The main reason for this is exactly the fact that the parasite wrapping is stronger in those regions due to a favorable local curvature and therefore, a lower energy penalty for membrane wrapping.

We have included an additional figure into the revised manuscript to discuss this point.

7) Similarly, is it possible to speculate on the influence of the merozoite shape on achieving alignment? For example, as well as providing an obvious advantage for initiating invasion, the egg shape means that the apex is titled towards the membrane in the most-likely configuration with maximum contact area (i.e. the peak of the distribution in Figure 3B is well above θ = π/2). This is not the case for a spherical shape, for which there is no preferred orientation, so that the merozoite may become more easily arrested with its apex pointing away from the RBC. Moreover, the tapering near the apex means there is naturally less material in its vicinity, so that alignment potentially could be achieved with an apex angle θ further from π.

We agree with the reviewer that this is a very interesting question, and clearly parasite shape should make a difference. One of the possible advantages of the egg shape is that it favors a preferred orientation, where the apex is relatively close to the membrane; another possibility is that the subsequent entry into the cell is facilitated by the pointed shape. However, the motion toward the successful alignment has some energy penalty and therefore, the probability of aligned configuration is smaller than that of the preferred orientation. For a spherical shape, there is no preferred orientation and the rotational motion does not have a significant energy penalty, which could potentially result in many stochastic trajectories to be far from the aligned configuration. Furthermore, for the same effective adhesion strength, the number of bonds can be different for different parasite shapes, which would affect the average displacement. As a result, all these contributions need to be carefully considered, and currently different hypotheses for the effect of different shapes on parasite alignment are too speculative and need a more thorough investigation. We are planning to study the effect of parasite shape on its alignment, but we prefer to avoid any speculations about this issue in the current manuscript.

8) Figure 4B: Why are there so many instances of very short alignment times, below the lower bound of 7 seconds observed experimentally by Yahata et al., 2012? Is this due to an effect being neglected or overestimated, or can the discrepancy be improved simply by tightening the alignment criteria (e.g. requiring θ > 0.9π rather than 0.8π)?

The range between 7 s and 44 s for parasite alignment from Yahata et al., 2012, is based on 10 experimental observations. So, the discrepancy can be just due to the lack of statistics of experimental data. Furthermore, our model is calibrated by other experiments (Weiss et al., 2015). We have tried to tighten the criteria, as suggested by the reviewer, but it seems not to affect the distribution of alignment times, it just reduces the total number of successful alignment events. Therefore, further experiments and potentially model improvements are needed to clarify whether there is a discrepancy and/or some effects are neglected or overestimated.

We have added a short discussion of this issue to the revised manuscript.

9) One concern is the sensitivity of the results to the discretisation used. This is because alignment requires that the distance between the merozoite apex and RBC membrane is very small and may become comparable to the discretisation length of the RBC membrane. The alignment criteria also necessitate examining small changes in the apex angle θ from π. For example, the change in the angle of the normal vector to the RBC membrane from one triangular face to the next scales as l/R, where l is the characteristic size of the triangles and R is the typical radius of curvature of the membrane. For the change in angle to be small (so that discretisation effects are negligible) requires l << R. However, the schematic in Figure 3A suggests that if the discretisation length is halved so that each segment becomes two segments, the value of θ could easily change by an amount comparable to the tolerance in the alignment criterion, i.e. 0.2π. Could the authors comment on this?

We completely agree with the reviewer that the discretization length affects the tightness of the alignment criteria. Note that the schematic in Figure 3A is not up to scale. In our simulations, the average discretization length of the RBC membrane is about l_0_=0.2 µm. The half circumference length of a parasite (corresponding to angle π) is πR, which is equal to about 12 l_0_ for R=0.75 µm, such that our angle resolution with respect to the parasite size is 0.1π. Therefore, we use 0.2π for the alignment criteria, which is large enough to avoid strong discretization effects. Simulations with a finer discretization are possible, but they become very expensive computationally.